# Controllable Generative Modeling via Causal Reasoning

**Avishek Joey Bose**                                    *joey.bose@mail.mcgill.ca*
*McGill University and Mila*

**Ricardo Pio Monti**                                    *rpmonti@meta.com*
*Meta*

**Aditya Grover**                                        *adityag@cs.ucla.edu*
*UCLA*

**Reviewed on OpenReview:** *https://openreview.net/forum?id=S8QISS_L9e5*

## Abstract

Deep latent variable generative models excel at generating complex, high-dimensional data, often exhibiting impressive generalization beyond the training distribution. However, many such models in use today are black-boxes trained on large unlabelled datasets with statistical objectives and lack an interpretable understanding of the latent space required for controlling the generative process. We propose CAGE, a framework for controllable generation in latent variable models based on causal reasoning. Given a pair of attributes, CAGE infers the implicit cause-effect relationships between these attributes as induced by a deep generative model. This is achieved by defining and estimating a novel notion of unit-level causal effects in the latent space of the generative model. Thereafter, we use the inferred cause-effect relationships to design a novel strategy for controllable generation based on counterfactual sampling. Through a series of large-scale synthetic and human evaluations, we demonstrate that generating counterfactual samples which respect the underlying causal relationships inferred via CAGE leads to subjectively more realistic images.

## 1 Introduction

Data generation using generative models is one of the fastest growing usecases of machine learning, with success stories across the artificial intelligence spectrum, from vision (Karras et al., 2020) and language (Brown et al., 2020) to scientific discovery (Wu et al., 2021) and sequential decision making (Chen et al., 2021). However, translating these successes into deployable applications in high-stake domains requires an interpretable understanding of the generative process for controllable generation.

Causality presents a natural framework to study and understand any generative process (Xu et al., 2020), and as a result, we believe it presents an a crucial ingredient for improving controllability of generative models. Specifically, we are interested in using causality for interpreting and controlling pretrained deep latent variable generative models (DLVGM), such as generative adversarial networks (GANs) Goodfellow et al. (2014) and variational autoencoders (VAEs) Kingma & Welling (2013). Prior work in controllable generation for such models can be categorized as either augmenting the loss function with disentanglement objectives, or post-hoc control approaches based on latent space explorations. The former class of methods e.g., Chen et al. (2016); Higgins et al. (2016) often introduce undesirable trade-offs with the standard generative modeling objectives (Locatello et al., 2019) and moreover, cannot be readily applied to the fast-growing zoo of state-of-the-art pretrained models in use today by the very virtue of needing to alter the original training methodology.

We focus on post-hoc control of DLVGMs, where control is specified w.r.t. a small set of interpretable meta-attributes of interest, such as gender or hair color for human faces. Here, the predominant approach for controllable generation is to identify directions of variation in the latent space for each meta-attribute and

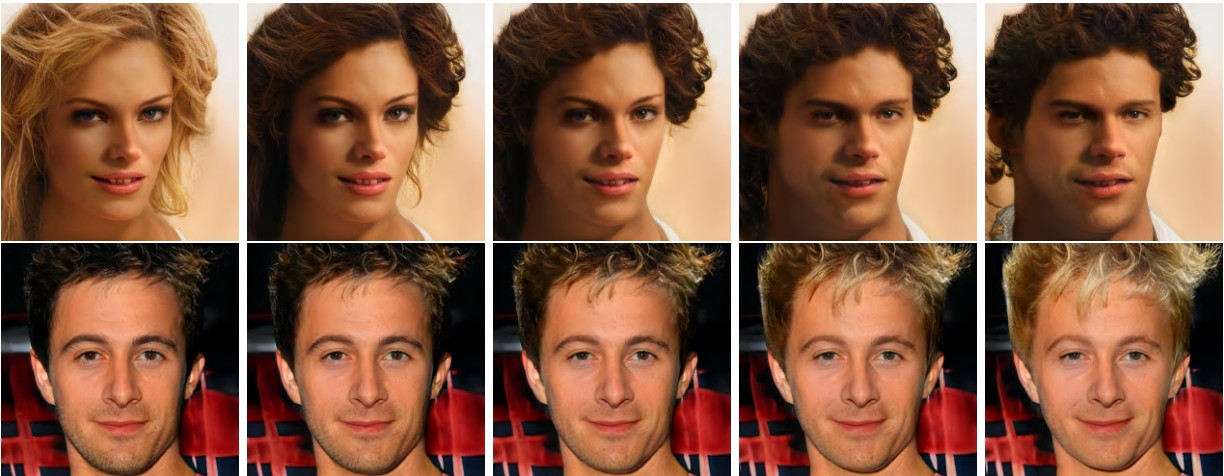

Figure 1: Counterfactual generation of Blonde Males using a deep generative model (Karras et al., 2020). CAGE infers the implicit causal structure as Gender → Blonde Hair. **Top:** The left panel contains the original image of a blond female, other panels display counterfactual samples where gender (treatment) is changed to male. **Bottom:** Equivalent visualizations starting from a non-blond male (left panel) and counterfactually changing hair color (treatment) to blond. Generating counterfactuals that respect the implicit causal structure yields subjectively better samples. This corresponds to the same-destination setting and is further outlined and empirically investigated in §4.1.

manipulate the latent code for an input image along these directions to achieve the desired control, e.g., Shen et al. (2020); Härkönen et al. (2020); Voynov & Babenko (2020); Khrulkov et al. (2021). While this procedure may work well in practice, it sidesteps a more fundamental question: *what are the interactions between these attributes in the latent space of a deep generative model?* Certainly, real world data generation processes have rich interplay—that go beyond mere correlation—between such meta-attributes. As a result, in practical scenarios, we do not expect that the generative model perceives these meta-attributes as independent and hence, inferring the cause-effect relationships between these meta-attributes can enhance our understanding and controllability of deep generative models.

We propose CAGE, a framework for inferring implicit cause-effect relationships in deep generative models. Our framework in inspired from the potential outcomes framework of the Neyman-Rubin causal model (Neyman, 1923; Rubin, 1974) and defines a new notion of *generative* average treatment effects (GATE). A fundamental problem of causal inference is that, by construction, we cannot observe the potential outcomes under all treatments (Holland, 1986) i.e., at any given time, any individual can be assigned only one treatment (a.k.a. the factual) but not both. However, when studying treatment effects for deep generative models, CAGE exploits the generative nature of such models to overcome this challenge and explicitly generate the counterfactual via a standard latent space manipulation strategy. Thereafter, we use an outcome attribute classifier to quantify the difference in outcomes for the factual and counterfactual generations and thus, estimate GATE. We further leverage the generative treatment effects to define a natural measure of causal direction over a pair of meta-attributes (e.g., gender and hair color) associated with a deep generative model. We refer to our overall framework for causal probing of deep generative models as CAGE.

We study the consistency of the inferred causal directions for two high-dimensional image datasets, MorphoMNIST (Pawlowski et al., 2020) and CelebAHQ (Karras et al., 2017), with known or biologically guessed prior cause-effect relationships. Finally, we use these inferred causal directions for augmenting post-hoc strategies for controllable generation. Specifically, we consider two scenarios for controllable generation: (a) *source manipulation*, wherein we wish to select which attribute of an input image should be manipulated for maximizing image fidelity, and (b) *destination manipulation*, where we wish to select input images that can be best manipulated to satisfy a target control attribute. We show that knowledge of the causal directions inferred via CAGE significantly improves the generation quality over baseline controllable generation strategies on the CelebAHQ dataset. In particular, we observe average improvements of 66.7% and 83.3% on source and destination manipulation scenarios respectively as measured by extensive human evaluations.

## 1.1 Related Work and Contributions

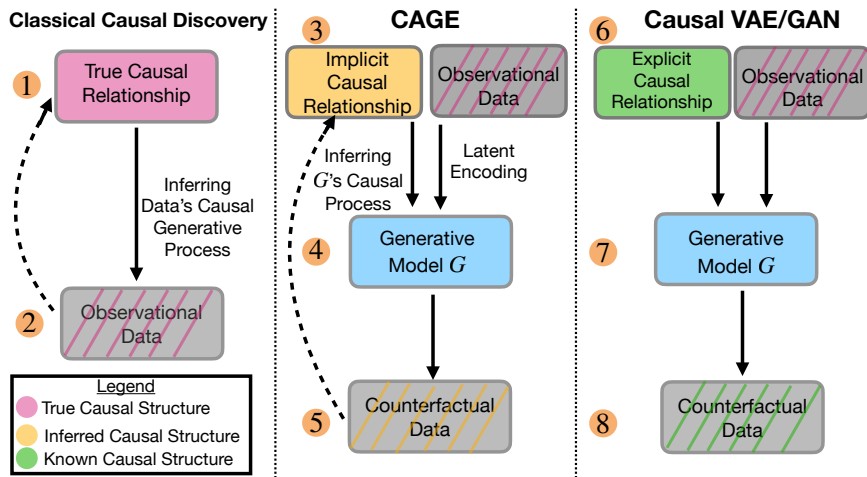

Figure 2: A visual summary of our proposed framework (middle panel) and how it relates to traditional causal discovery (left panel) and Causal VAE/GAN (right panel) respectively. Traditional causal discovery uses observational data to infer causal relationships over the true data generating mechanism. In contrast, the goal of CAGE is to infer implicit causal relationships within a deep generative model. We note that the implicit causal relationships within a generator **need not** match the causal relationships over the data used to train such a model. Finally, methods such as Causal VAE/GAN train generative models under a prescribed causal structure, whereas the goal of our proposed framework is to reliably uncover implicit causal structure over any generative model.

Our proposed framework lies at the intersection of causal inference and deep generative modeling. While CAGE borrows important concepts and ideas from each field, we also make a series of distinctive contributions to each as outlined in Figure 2.

Traditional causal discovery methods are depicted in the left panel of Figure 2. These methods operate over observational data and seek to infer the underlying causal relationship which generated the data. The intrinsic appeal of causal discovery methods is that they are able to infer the latent causal structure of complex datasets, thus providing an explicit description of the underlying generative mechanism. However, in order to achieve this, a series of (potentially restrictive) assumptions are typically required, such as assuming causal relationships are linear (Shimizu et al., 2006; Hyvärinen & Smith, 2013; Peters & Bühlmann, 2014; Peters et al., 2016) or follow tightly parameterized non-linear relationships such as the additive noise model of Hoyer et al. (2008). CAGE, depicted in the middle panel, borrows ideas from causal discovery in order to infer latent causal structure within a deep generative model, $G$. This is fundamentally different to learning the causal structure of the data itself, which is the goal of causal discovery. Specifically, this is because the implicit causal structure within a deep generative model will only respect the true causal structure of the data under strict conditions (Klindt et al., 2020; Khemakhem et al., 2020; Gresele et al., 2021; Monti et al., 2020), which do not apply for a wide range of popular deep generative models, for example many state-of-the-art GANs or Normalizing Flows. In the absence of such guarantees, the implicit causal structure within $G$ can be arbitrarily different from the causal relationships which generated the observed data.

Finally, the right panel of Figure 2 highlights a related avenue of research which aims to train deep generative models under a prescribed causal structure. Examples include CausalGAN, CausalVAE, and counterfactual generative networks (Kocaoglu et al., 2017; Yang et al., 2021; Sauer & Geiger, 2021). Despite sharing some similarities, these methods are based on instilling a given causal structure within a generative model, $G$, by changing the objective function. In stark contrast, the objective of CAGE is to infer such causal structure over any generator, $G$ *without* introducing any additional constraints or trade-offs whilst training $G$ since we only look to understand the implicit causal associations *after* training. As such, CAGE is applicable to any deep generative model, and we provide examples with VAEs, Normalizing Flows, and GANs in this work.

## 2 Background and Preliminaries

### 2.1 The potential outcomes framework

We briefly overview the potential outcomes framework of Neyman (1923) and Rubin (1974), upon which we base our proposed method in the next section. In its simplest form, this framework considers the causal effect of assigning a binary treatment, $T \in \{0, 1\}$. Such a framework posits the existence of potential outcomes for the $i$th individual both whilst receiving treatment, $Y_{T=1}(i)$, and when treatment is withheld, $Y_{T=0}(i)$. The causal effect for a given individual, $i$, is defined as the difference in these potential outcomes.

The "fundamental problem of causal inference" is that whilst we may posit the existence of both potential outcomes, $Y_{T=1}(i)$ and $Y_{T=0}(i)$, we only ever observe the outcome under one treatment (Holland, 1986). For this reason, causal inference is often performed at the population level, by considering quantities such as the average treatment effect (ATE):

$$\tau = \mathbb{E}_i \left[ Y_{T=1}(i) - Y_{T=0}(i) \right]. \tag{1}$$

Equation (1) is the expected difference in potential outcomes of individuals receiving treatment $T = 1$ and $T = 0$. Under standard assumptions of unconfoundedness, positivity, and stable unit treatment value assumption, the ATE is identifiable and can be estimated by a statistical estimate of the associational difference in expectations (Rubin, 1978). Please see Appendix §A for a light introduction to causal inference as it pertains to our framework CAGE.

### 2.2 Deep latent variable generative models

The goal of generative modeling is to learn an underlying data distribution given a training dataset. For high-dimensional data, this can be done effectively via deep neural networks by learning a mapping $G$ from unobserved factors of variation $\mathcal{Z}$ (i.e., latent variables) to observed variables $\mathcal{X}$ (e.g., raw pixels). The mapping $G$ is parameterized using deep neural networks, and can be learned via a variety of training objectives, such as adversarial training (e.g., generative adversarial networks (GAN; Goodfellow et al. (2014)), exact maximum likelihood estimation (e.g., Normalizing Flows (Dinh et al., 2017)), and variational Bayes (e.g., Variational Autoencoders (VAE; Kingma & Welling (2013); Rezende & Mohamed (2015)). For our current work, we can use any such latent variable generative model. Our experiments will exhibit this generality by considering powerful families of flows (Song et al., 2019) and GANs (Karras et al., 2020).

In addition to $G$ (i.e., the decoder), we will assume the existence of an encoder $\text{Enc} : \mathcal{X} \to \mathcal{Z}$ that can map any observed datapoint to its latent vector representation. For instance, for flows, the encoder is given by the inverse of the generator mapping, $\text{Enc} = G^{-1}$. For VAEs, the encoder is the variational posterior distribution trained alongside $G$. For GANs, either the encoder can be trained during training (e.g., Donahue et al. (2016); Dumoulin et al. (2016)), after training (e.g., Richardson et al. (2021)), or implicitly learned by backpropagating through $G$ to find a latent vector $z \in \mathcal{Z}$ that best represents a target input $x$, as measured via a reconstruction loss (e.g., Lipton & Tripathi (2017); Bora et al. (2017)) or a pretrained domain classifier (e.g., (Karras et al., 2020)).

## 3 Causally aware controllable generation

In this work we approach the challenge of controllable generation from the perspective of causality. In particular, we posit the existence of implicit causal structure within a deep generative model, $G$. We study this structure over metadata attributes, for example gender and hair color as described in Figure 1, with the underlying motivation that an effect attribute is by construction leads to more fine-grained control of $G$ whilst the converse is true for causal attributes. For example, a deep generative model may internalize that changing the meta-attribute gender may causally materialize the meta-attribute pertaining to the presence or absence of a mustache, but not vice-versa. Our goal is therefore to understand the implicit causal structure—if any—encoded within a deep generative model by asking the following question:

*In the context of a generator, $G$, does considering the counterfactual over a given "causal" attribute (e.g., gender) lead to significant measurable changes in an "effect" attribute (e.g., hair color)?*

Formally, we assume full (whitebox) access to a pretrained latent variable deep generative model $G : \mathcal{Z} \to \mathcal{X}$. We further assume a finite dataset of $n$ annotated observations, $D = \bigcup_{i=1}^{n} \{\mathbf{x}_D(i), m_1(i), m_2(i)\}$. Here, the examples $\{\mathbf{x}_D(1) \ldots, \mathbf{x}_D(N)\}$ are drawn from the model's training distribution and we annotate each example $\mathbf{x}_D(i)$ with binary metadata for two variables $m_1(i), m_2(i) \in \{0, 1\}$. These two annotated variables can be drawn from a larger *unobserved* (w.r.t. $G$) causal process. For example, $\mathbf{x}_D(i)$ could correspond to a high-dimensional image and $m_1(i)$ and $m_2(i)$ could be any two attributes of the image, such as gender expression and hair color in Figure 1. Apriori, we do not know which of the variables (if any) is cause or effect but seek to identify their relationship within $G$.

We note that any causal conclusions obtained under our proposed framework only reflect implicit properties of a generator $G$, as opposed to properties of the data on which the generator was trained. That is to say, it is perfectly permitted, for $G$ to have learned a causal relationship that is inconsistent with the observational data used for training. As we shall see in §4, shedding light on the implicit causal relationship learned by $G$ is abundantly useful for the purposes of controllable generation. As a result, this work—by design—is in sharp contrast with the majority of causal inference literature that focuses on identifying the generative mechanisms of a fixed dataset as opposed to our focus on the generative model itself.

## 3.1 Overview of the CAGE framework

We propose to extend the potential outcomes framework described in §2.1 to accommodate DLVGMs. Our approach is premised on computing a *generative* average treatment effect for a candidate causal attribute, $m_c$, on a given effect attribute, $m_e$, for a given generator, $G$. Unlike traditional approaches to treatment effect estimation, the key observation in CAGE is that we can simulate the counterfactuals under a given treatment using the generative model $G$. In practice, this amounts to traversing the latent space of $G$ using pre-specified strategy. For instance, we may move orthogonally to learned meta-attribute hyperplanes or even consider a more exotic non-linear latent manipulation strategy. It is important to re-emphasize here the distinction between between simulating counterfactuals with respect to the true causal relationships in the data versus CAGE which uses the generative models own latent space to perform counterfactuals queries. In order to investigate the cause-effect relationship between two meta-attributes—e.g. gender and hair color, let us initially arbitrarily assign gender to be the treatment $m_1 = m_c$ (cause) and hair color to be the outcome $m_2 = m_e$ (effect). We note this choice is arbitrary and the overall algorithm will also consider the reverse assignment (i.e., $m_1 = m_c$ and $m_2 = m_e$) and if needed, also reveal independence between the variables via a statistical test. Then we may simulate the effects of treatment assignments by manipulating the gender attribute in the latent space and measuring the corresponding change in outcomes. We next discuss our strategies for counterfactual manipulation. The overall pseudocode for CAGE is summarized in Algorithm 1.

## 3.2 Simulating Treatments via Latent Counterfactual Manipulations

In order to simulate the assignment of counterfactuals, we follow a three-step procedure.

**Step 1:** Our first step is to train a latent space classifier for the attributes using $D$. Here, we use the encoder to first project every input $\mathbf{x}_D(i)$ in $D$ to its latent encoding $\mathbf{z}_D(i) = \text{ENC}(\mathbf{x}_D(i))$.[1] Given the latent encodings $\mathbf{z}_D(i)$ and their annotations for $m_c$ and $m_e$ attributes, we train a (probabilistic) classifiers, $\phi_c, \phi_e : \mathcal{Z} \to [0, 1]$, to discriminate between binary causal and effect attribute values given latent representations.

**Step 2:** Since we are interested in casual discovery for the generative model, in the second step, we use $G$ to create a dataset of annotated examples by sampling $k$ latent vectors $Z = \bigcup_{i=1}^{k} \{\mathbf{z}(i)\}$ from the prior of the generative model. Let $X = \bigcup_{i=1}^{k} \{\mathbf{x}(i) = G(\mathbf{z}(i))\}$ denote the corresponding generations. Since sampling latents from the prior generative model are typically inexpensive, the size of our generated dataset can be fairly large. For each generated example $\mathbf{x}(i) \in X$, we obtain its factual treatment by simply using the latent space classifier as $m_c(i) = \mathbb{1}[\phi_c(\mathbf{z}(i)) > 0]$.

**Step 3:** For a generated sample, $\mathbf{x}(i) \in X$, assume without loss of generality, that $m_c = 0$ is the factual treatment obtained via Step 2. We can derive analogous expressions for factual treatments $m_c = 1$, but we skip those for brevity. Finally, we define the counterfactual latent with respect to setting the treatment

---

[1]For stochastic encoders such as in VAEs, we consider the mean of the encoding distribution.

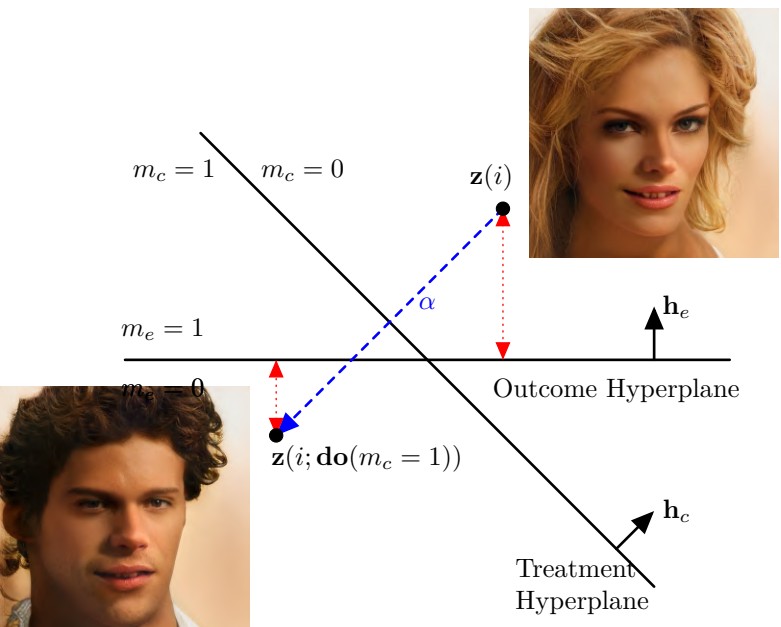

Figure 3: Linear Counterfactual Manipulation in the Latent Space of Generative Models in CAGE.

attribute $m_c = 1$ as using the **do** operator as:

$$\mathbf{z}(i; \mathbf{do}(m_c = 1)) = \pi_c(\mathbf{z}(i), \alpha), \tag{2}$$

where $\pi_c$ a latent space interpolation strategy $\alpha \in \mathbb{R}_+$ is a hyperparameter which is positive scalar that controls the extent to which we manipulate. Note, that $\pi_c$ itself is very general and it could be as simple as linear interpolation if the latent space is linearly separable or a more exotic approach that exploits the known geometric structure of the latent space. A counterfactual sample can subsequently be obtained by simply pushing forward the counterfactual latent through $G$:

$$\mathbf{x}(i; \mathbf{do}(m_c = 1)) = G(\mathbf{z}(i; \mathbf{do}(m_c = 1))). \tag{3}$$

### 3.3 Case Study: Linear Latent Counterfactuals

As a concrete example, we consider instantiating our CAGE with counterfactuals crafted by linearly interpolating across a linear decision boundary. We outline below how such latent linear counterfactuals are materialized in the following steps:

**Step 1:** We first train a probabilistic linear classifiers $\phi_c, \phi_e : \mathcal{Z} \to [0, 1]$ on $D$ to minimize the cross-entropy loss and thus, discriminate between binary causal and effect attribute values given latent encodings $\mathbf{z}_D(i)$ and their annotations for $m_c$ and $m_e$. Note, that we will use $\phi_e$ in §3.4 quantify treatment effects. Concretely, by restricting ourselves to linear classifiers we obtain a unit hyperplane that encodes the classification boundary. We use $\mathbf{h}_c$ to denote the normal vector to the hyperplane for the meta-attribute $m_c$.

**Step 2:** After generating our dataset $X = \bigcup_{i=1}^{k} \{\mathbf{x}(i) = G(\mathbf{z}(i))\}$ using $G$ we can freely and cheaply apply our trained latent linear classifier to obtain factual treatments for each generated sample $\mathbf{x}(i) \in X$ as $m_c(i) = \mathbb{1}[\phi_c(\mathbf{z}(i)) > 0]$.

**Step 3:** As outlined in the general CAGE framework for a generated sample, $\mathbf{x}(i) \in X$ we denote $m_c = 0$ as the factual treatment obtained via Step 2. and we define the counterfactual latent—i.e. $m_c = 1$—using the **do** operator as:

$$\mathbf{z}(i; \mathbf{do}(m_c = 1)) = \mathbf{z}(i) + \alpha \mathbf{h}_c, \tag{4}$$

where $\pi_c = \mathbf{z}(i) + \alpha \mathbf{h}_c$. Equation 4 corresponds to moving linearly along the hyperplane normal, $\mathbf{h}_c$, which encodes attribute $m_c$ for simulating the counterfactual. Such an approach is premised on the assumption that

$\mathcal{Z}$ is linearly separable with respect to a semantically meaningful latent attribute, which was first observed and empirically validated for GANs (Denton et al., 2019). A counterfactual sample can subsequently be obtained by simply pushing forward the counterfactual latent through $G$:

$$\mathbf{x}(i; \mathbf{do}(m_c = 1)) = G(\mathbf{z}(i; \mathbf{do}(m_c = 1))). \tag{5}$$

Figure 3 illustrates such a linear counterfactual manipulation strategy.

### 3.4 Quantifying Treatment Effects via Proxy Classifiers

We now prescribe an estimate to quantify treatment effects using our trained latent classifiers. At a high level we seek to measure the difference in treatment effects when manipulating one meta-attribute, $m_c$, on the other meta-attribute $m_e$ via simulating counterfactuals. By considering both directions $m_c \rightarrow m_e$ and $m_e \rightarrow m_c$ and leveraging the fact that causal treatments leave measurable imprints on effect variables and not vice-versa we can assign an approximate measure of the causal relationship by considering the difference between the two directions at a population level.

In order to estimate treatment effects—e.g. $m_c \rightarrow m_e$—we require an estimate of the presence or absence of the effect attribute, $m_e$, for the counterfactuals. To this end, we propose the use of binary classifiers trained to detect the presence of an effect variable. Intuitively, using Equation 5 and the corresponding geometric manipulations required to obtain interventional samples $\mathbf{x}(i; \mathbf{do}(m_c = 1))$ from a generative model, $G$ we can simulate the impact of prescribing a treatment to an individual. We dub this the *generative* individual treatment effect (GITE) estimate $\tau_{\text{GITE}}$ which is defined as:

$$\tau_{\text{GITE}}(m_c \rightarrow m_e) = m_e(i; \mathbf{do}(m_c = 1)) - m_e(i; \mathbf{do}(m_c = 0)). \tag{6}$$

Similar to the ATE estimate eqn. 1 we can define a population level estimate by taking an expectation over the generated examples. We term this *generative* average treatment effect (GATE) of $m_c$ on $m_e$ defined as:

$$\tau_{\text{GATE}}(m_c \rightarrow m_e) = \mathbb{E}_i[m_e(i; \mathbf{do}(m_c = 1)) - m_e(i; \mathbf{do}(m_c = 0))] \tag{7}$$

To materially compute treatments effects required in equations 6 and 7—e.g. $m_e(i; \mathbf{do}(m_c = 1))$, we propose two candidate options that directly act on the counterfactual latent:

(1) Latent space classifiers are trained in the latent space to detect the presence or absence of the effect variable in the counterfactual latent as:

$$m_e(i; \mathbf{do}(m_c = 1)) = \phi_e\left(\mathbf{z}(i; \mathbf{do}(m_c = 1))\right). \tag{8}$$

(2) Observed space classifiers are trained in the observation space and perform classifications after generating the samples corresponding to the counterfactual latent as:

$$m_e(i; \mathbf{do}(m_c = 1)) = \psi_e\left(\mathbf{x}(i; \mathbf{do}(m_c = 1))\right). \tag{9}$$

As we can see from Equation 7, our GATE estimate differs from the conventional ATE estimates in Equation 1 in the use of generative models and classifiers to explicitly generate the counterfactual. For reliable estimation, it is reasonable to expect some standard assumptions from causality to apply to even our setting, such as unconfoundedness, positivity, and SUTVA. In addition to these standard assumptions, the quality of the generative model and classifiers play an important role. In §3.3, for the generative model, we designed our counterfactual manipulation scheme assuming that the latent space is linearly separable in the attributes of interest. Further, implicit in our framework is the assumption that the generative model can generalize outside the training set to include the support of the counterfactual distribution which is needed for Equation 8 to provide a reliable effect estimate of the counterfactual. On the use of classifiers, we need accurate and calibrated classifiers (ideally, Bayes optimal) for both manipulating latent vectors in Equation 4 and quantifying generative average treatment effects in Equation 7. While it is impossible to test in—full veracity— the above assumptions on real-world distributions, there is significant empirical evidence in the last few years that suggest that modern deep generative models and classifiers can indeed satisfy the above

requirements for many practical usecases (Denton et al., 2019; Brown et al., 2020). The theoretical properties and these necessary assumptions of CAGE are further discussed in §B.

$\Delta\tau$ **as a measure of causal direction.** Finally, we note that the preceding sections have focused on the challenge of quantifying if a variable, $m_c$, has a causal effect on second variable, $m_e$. This assumes knowledge of the causal ordering over variables As such, we can extend our approach to define a measure of causal direction over any pair of variables by considering the difference in absolute GATE scores when either variable is considered as the treatment:

$$\Delta\tau = |\tau_{\text{GATE}}(m_c \rightarrow m_e)| - |\tau_{\text{GATE}}(m_e \rightarrow m_c)| \tag{10}$$

The $\Delta\tau$ score determines the magnitude of the difference in outcomes when either attribute is prescribed as the treatment. Intuitively, when using the true causal attribute, according to $G$, as the treatment we expect a signature of the interventions to be large in magnitude. Conversely, the magnitude of disturbance measured when manipulating the true effect variable should be markedly smaller. Thus $\Delta\tau > 0$ corroborates that the chosen causal ordering is indeed the one supported by $G$, while a $\Delta\tau < 0$ implies that the chosen causal ordering may in fact be reversed.

**A null distribution for treatment effects.** A further important consideration relates to how we might determine whether an estimated GATE is statistically significant. To address this, we can obtain an empirical sample of GATE scores under the null hypothesis where the attribute $m_c$ has no causal association with $m_e$. A simple manner through which this can be obtained is via randomization and permutation testing. We can randomly shuffle the values of a causal attribute, $m_c$, thereby removing any potential causal association. This corresponds to performing interventions over the latent space of $G$ which are effectively random projections. This process is repeated many times to obtain an empirical distribution for a GATE under the null hypothesis. Given an empirical distribution of GATEs under the null, we can obtain a $p$-value for our observed GATE.

---

**Algorithm 1** Obtaining CAGE scores $\Delta\tau$

---

**Inputs:** Generative model $G$, Dataset $D$, Attributes $m_1, m_2$, Counterfactual data size $k$

1: $m_c \leftarrow [m_1]$, $m_e \leftarrow [m_2]$ {// Randomly Assign meta-attributes as cause $m_c$ and effect $m_e$}
2: **for** $(\mathbf{x}_D(i), m_c(i), m_e(i))$ in $D$ **do**
3:     $\mathbf{z}_D(i) = \text{ENC}(\mathbf{x}_D(i))$
4:     $D_z = D_z \cup \{\mathbf{z}_D(i), m_c(i), m_e(i)\}$
5: **end for**
6: $\phi_c \leftarrow \mathbb{E}_{z, m_c(i) \sim D_z}[\mathcal{L}(z, m_c(i); \phi_c)]$ {// Train probabilistic meta-attribute classifier $\phi_c$}
7: $\phi_e \leftarrow \mathbb{E}_{z, m_e(i) \sim D_z}[\mathcal{L}(z, m_e(i); \phi_e)]$ {// Train probabilistic meta-attribute classifier $\phi_e$}
8: **for** $i \in [k]$ **do**
9:     $Z = Z \cup z(i) \sim \mathcal{N}(0, I)$ {// Sampling from $G$'s prior}
10:     $X = X \cup \{x(i) = G(z(i))\}$
11: **end for**
12: **for** $z(i) \in Z$ **do**
13:     $\mathbf{z}(i; \mathbf{do}(m_c = 1)) = \pi_c(\mathbf{z}(i), \alpha)$ {// Counterfactual latent by applying the treatment $m_c = 1$}
14:     $\mathbf{z}(i; \mathbf{do}(m_c = 0)) = \pi_c(\mathbf{z}(i), -\alpha)$ {// Counterfactual latent by applying the treatment $m_c = 0$}
15:     $\mathbf{z}(i; \mathbf{do}(m_e = 1)) = \pi_e(\mathbf{z}(i), \alpha)$ {// Counterfactual latent by applying the treatment $m_e = 1$}
16:     $\mathbf{z}(i; \mathbf{do}(m_e = 0)) = \pi_e(\mathbf{z}(i), -\alpha)$ {// Counterfactual latent by applying the treatment $m_e = 0$}
17:     $m_e(\mathbf{do}(m_c = 1)) = m_e(\mathbf{do}(m_c = 1)) \bigcup \{m_e(i; \mathbf{do}(m_c = 1)) = \phi_e(\mathbf{z}(i; \mathbf{do}(m_c = 1)))\}$
18:     $m_e(\mathbf{do}(m_c = 0)) = m_e(\mathbf{do}(m_c = 1)) \bigcup \{m_e(i; \mathbf{do}(m_c = 0)) = \phi_e(\mathbf{z}(i; \mathbf{do}(m_c = 0)))\}$
19:     $m_c(\mathbf{do}(m_e = 1)) = m_c(\mathbf{do}(m_e = 1)) \bigcup \{m_c(i; \mathbf{do}(m_e = 1)) = \phi_c(\mathbf{z}(i; \mathbf{do}(m_e = 1)))\}$
20:     $m_c(\mathbf{do}(m_e = 0)) = m_c(\mathbf{do}(m_e = 0)) \bigcup \{m_c(i; \mathbf{do}(m_e = 0))\phi_c(\mathbf{z}(i; \mathbf{do}(m_e = 0)))\}$
21: **end for**
22: $\tau_{\text{GATE}}(m_c \rightarrow m_e) = \mathbb{E}_i[m_e(i; \mathbf{do}(m_c = 1)) - m_e(i; \mathbf{do}(m_c = 0))]$
23: $\tau_{\text{GATE}}(m_e \rightarrow m_c) = \mathbb{E}_i[m_c(i; \mathbf{do}(m_e = 1)) - m_c(i; \mathbf{do}(m_e = 0))]$
24: $\Delta\tau = |\tau_{\text{GATE}}(m_c \rightarrow m_e)| - |\tau_{\text{GATE}}(m_e \rightarrow m_c)|$
25: **return:** $\Delta\tau$ =0

---

## 4 Experiments

We evaluate the ability of CAGE towards inferring causal relationships in pretrained deep generative models on both synthetic and high-dimensional datasets such as high-resolution images of faces. As outlined in §3.4 the estimate for the causal association uncovered by CAGE is predicated on a few key assumptions. Our experiments section is organized as follows: we first test the empirical caliber of the causal estimate returned by CAGE by designing a series of unit tests that explicitly probe each testable assumption. Concretely, through our experiments we seek to answer the following questions:

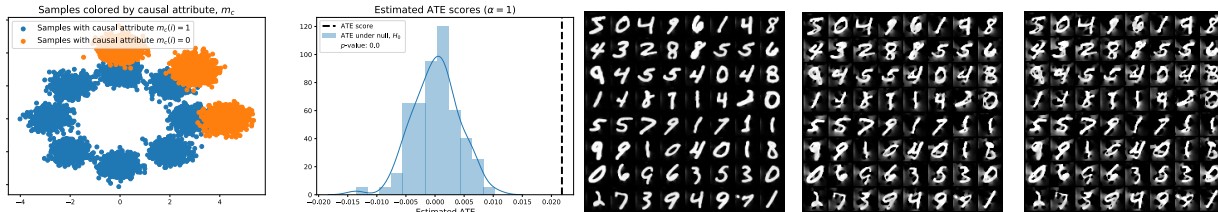

Figure 4: **Left-1**: A scatterplot of MoG with points colored by the value of causal attribute, $m_c$. **Left-2**: Histogram of GATE values under null hypothesis, $H_0$, and vertical line denoting estimated GATE. **Left-3** Generating counterfactuals of MorphoMNIST digits by reducing the thickness of each digit without changing the average intensity. **Left-4,5** Progressively decreasing digit thickness leads to background pixels being illuminated as the digits get thinner.

- **(Q1) Generator Causal Discovery:** Can CAGE recover the true causal structure on synthetic datasets?
- **(Q2) Linear Separability:** To what degree is the latent space linearly separable in meta-attributes of interest?
- **(Q3) Statistical Significance:** How statistically significant is GATE under the null distribution?

Our empirical result summarizing these core experimental questions is presented in table 3. Next, we turn to our main experimental findings in §4.1 on controllable generation leveraging causal insights supplied by CAGE in human studies under two settings that we term as same source or same destination manipulation.

### Q1. Generator Causal Discovery

**Mixture of Gaussians**. We consider two synthetic datasets with known causal relationships within the data. In our first experiment we consider the following toy setup: data is generated according to two distinct mixture distributions as shown in the left panel of Figure 4 where each color denotes a mixture. We define the causal attribute, $m_c$, to be which mixture each sample is drawn from (i.e., from the mixture of 8 Gaussians, in blue, or the mixture of 3 Gaussians, in orange). We further define an effect variable, $m_e$, to be defined as one, if the mean of the $(x, y)$-coordinates is less than 2.5 and zero otherwise. In this manner, we can see that $m_c$ has a causal influence over $m_e$ by determining which mixture each sample is drawn from. We employ a Masked Autoregressive Flow (Papamakarios et al., 2017) as the deep generative model. The middle panel of Figure 4 visualises the distribution of $\tau_{\text{GATE}}$ under the null as well as the estimated GATE, $\tau_{\text{GATE}}(m_c \rightarrow m_e)$, which is significantly larger in magnitude while the right plots $\Delta \tau$ as a function of $\alpha$, positive values corroborate that $m_c \rightarrow m_e$.

**MorphoMNIST**. In our second synthetic experiment we evaluate the causal relationships learned by powerful deep normalizing flow models on a synthetic dataset based on MNIST dubbed MorphoMNIST (Pawlowski et al., 2020). Here, the original MNIST digits are modified to respect a causal structure whereby the stroke thickness of the digits is a cause to the brightness (i.e. $T \rightarrow I$). Specifically, thicker digits are brighter while thinner digits are dimmer under the prescribed causal graph. We train a powerful normalizing flow in Mintnet (Song et al., 2019) and interrogate the direction of causality—if any—between thickness and intensity by first projecting all test set samples to their corresponding latent vectors. Figure 4 shows qualitative examples of counterfactuals, where we observe that the generated counterfactuals reduce the thickness while maintaining the average intensity. Specifically, the model generates digits that contain holes but to

compensate for the drop in intensity the model instead increases the intensity of the background pixels. Such generations are in line with expectations as $\Delta\tau > 0$ (see §C.2) indicates that the model has learned that $T \to I$ and as such manipulating the causal variable propagates influence to the effect variable manifesting itself as an increase in background intensity.

### Q2. Linear Separability

We test the linear separability of latent classifiers on a powerful generative model StyleGAN2 (Karras et al., 2020) which is pretrained on the FlickrFacesHQ (Karras et al., 2019) but finetuned using 10% of CelebaHQ which contains annotated meta-attributes such as Gender and Haircolor for photo-realistic facial images. For latent linear classifiers we use an SVM based classifier as found in the widely used Sci-kit learn library (Pedregosa et al., 2011). In Table 1 we report the test accuracy of our latent SVM based classifier and observe that many attributes of interest exhibit high degrees of linear separability. However, certain attributes such as Pointy Nose and Brown Hair are less linearly separable and as a result the causal relationships inferred by CAGE are less reliable. In particular, meta-attributes such as Gender and Goatee are quite linearly separable while meta-attributes such Brown Hair and Pointy Nose are less linearly separable and any causal estimate returned by CAGE is less reliable.

### Q3. Statistical Significance

We now perform causal discovery over a pair of meta-attributes within the latent space of a pretrained StyleGAN2 $G$. In Table 3, we report the direction of causality found using $\Delta\tau$ on pairs that pass the linear separability unit test above. We observe that $\Delta\tau$ is consistently capable of diagnosing the causal relationship implicit to $G$. For example, for the causal relationship: Gender $\to$ Mustache, we observe counterfactual samples where gender (top row) is manipulated—i.e. from male to female–the mustache attribute also changes. In contrast, when the effect variable (bottom row) is manipulated there is a reduced impact on the perceived gender. For each pair, we also generate counterfactual samples by manipulating one attribute and qualitatively observing downstream effects on the other. To ascertain that our perceived scores are significant we plot a histogram $\Delta\tau$ scores under 100 random latent projections which allows us to reject the null hypothesis if the true $\Delta\tau$ (black line) is significant (e.g. $p$-val $< 0.05$).

**Correlation vs. Causation**. To determine the extent to which correlated attributes affect CAGE, we compute the Maximum Mean Discrepancy (MMD) metric (Gretton et al., 2012) between the latent conditional and counterfactual distributions as a function of the interpolant strength $\alpha$ in the final column in Table 3. Concretely, we filter the generated samples by their attribute values (as determined via a pretrained latent space classifier) as an estimate for the conditional $p(z|m_i = 1)$ and compare it with $p(z|\mathbf{do}(m_i = 1))$ obtained by generating counterfactuals. We find that in all cases the MMD is large, signifying that the conditional and counterfactual distributions are quite different. For example, the attributes Gender and Smiling are highly correlated in the CelebAHQ dataset (14.2% males vs. 32.8% females are smiling) and in the generated examples (10.0% males vs. 36.0% females are smiling, with labels determined by a pretrained classifier) but CAGE infers the two attributes to be independent as substantiated with the generated counterfactuals.

**Latent Causal Discovery Baselines**. As noted in Section 1.1, the primary objective of CAGE is to infer causal structuve over metadata attributes, $m_i$, encoded in a generator, $G$, as opposed to over observations, $\mathbf{X}$. As such, CAGE looks to solve a fundamentally different problem from traditional causal discovery methods. Despite these differences we nonetheless include a series of latent causal discovery baselines: 1. LiNGAM (Shimizu et al., 2006) 2. Additive Noise Model (Hoyer et al., 2008), and 3. DAGs with No Tears (Zhang et al., 2018). We adapt such baselines so as to be comparable with CAGE by having them operate on the class probabilities learned by the proxy classifiers described in §3.4. We note that this is required both for a meaningful comparison with CAGE, but also as a result of the poor scaling properties of many causal discovery algorithms which cannot be directly applied over image data. In general, we find little agreement among the baselines which is expected as they each make different assumptions and as a result this inhibits their easy application to downstream causally informed controllable generation.

### 4.1 Sample Quality of Counterfactual Generation

We now investigate the visual fidelity of counterfactual samples crafted by manipulating the causal variable or the effect variable as uncovered through CAGE in CelebaHQ.

**Experiment Design.** For each pair of meta-attributes, we consider two experimental setups: (1) same source and (2) same destination manipulation. In the same source setting, the starting point for generation is a source image (e.g., male with no mustache) which is then manipulated twice by the causal variable and the effect variable respectively to produce two different images (e.g., female without a mustache and male with a mustache). In the same destination setting, we reverse the previous process by attempting to arrive at the same set of meta-attributes for two different real starting images after manipulation. For example, let's say we wish to generate images of men without mustaches. Here, we may start from a female with no mustache and change its gender, or start from a male with a mustache and remove the mustache. An illustration of these settings can be found in Fig. 8. For each attribute pair and both of the above settings, we generate 100 pairs of images which are evaluated using human annotators solicited through Amazon Mechanical Turk to rate images on the basis of their realism. See §C for details and §E for example samples.

**Results.** We report our findings in Table 2 where we find in the same source setting generating counterfactuals by modifying the causal variable leads to subjectively better visual samples as determined by human annotators in all but two settings (e.g. Gender → Bald). Conversely, in the same destination setting modifying the effect variable is preferable to the causal one when evaluating for sample quality. These results highlight that depending on how $G$ is used downstream—e.g. a same source setting like data augmentation for rare classes a higher degree of visual fidelity can through controllable generation using a causal attribute. On the other hand, when doing conditional generation in tasks such as in-painting it is more desirable to modify the effect variable to produce higher quality visual samples.

| Attributes | SVM Classifier Accuracy |
|---|---|
| Gender | 90.0 |
| Mustache | 79.7 |
| Bald | 70.8 |
| Goatee | 83.2 |
| Rosy Cheeks | 72.8 |
| Age | 75.1 |
| Pointy Nose | 64.6 |
| Brown Hair | 65.4 |
| Smiling | 79.6 |
| Blond Hair | 84.8 |

Table 1: Latent Space SVM Classifier Test Accuracies on StyleGAN2 finetuned on CelebaHQ.

| CAGE direction | Same Source ↑ | Same Dest. ↓ |
|---|---|---|
| Gender → Mustache | 231/300* | 51/300* |
| Gender → Bald | 156/300 | 138/300 |
| Gender → Goatee | 254/300* | 91/300* |
| Age ← Bald | 281/300* | 56/300* |
| Gender → Rosy Cheeks | 183/300* | 97/300* |
| Gender → Blond Hair | 157/300 | 126/300* |

Table 2: Counterfactual Sample Quality Results. Each cell represents the percentage of annotators that preferred the counterfactual generated by modifying the causal variable in each pair of attributes. We add a * to denote statistical significance at level $\alpha = 0.05$, after Bonferroni correction.

## 5 Conclusion

We proposed CAGE, a framework for inferring cause-effect relationships within the latent space of deep generative model via geometric manipulations for causally guided controllable generation. CAGE is well suited to a wide family of modern deep generative models trained on complex high-dimensional data and does not require any altering of the original training objective nor hard to obtain counterfactual data. Empirically, we find CAGE reliably extracts correct cause-effect relationships in controlled settings like MorphoMNIST. On high-resolution image data such as CelebAHQ, CAGE reveals cause-effect relationships that are best supported by generated counterfactual samples. Finally, using causal insights supplied by CAGE we also perform a large scale human evaluation of controllable generation revealing that higher quality samples are generated by controlling the causal variable in same source settings while manipulating the effect variable leads to a better fine-grained generation in same destination settings.

## Acknowledgements

The authors would like to thank all anonymous reviewers for their feedback. In addition, the authors thank the workers of Amazon Mechanical Turk for assisting with the qualitative evaluation of generated samples. AJB performed this work during an internship at Meta AI. This work was supported by a Sony Faculty Innovation Award for AG. Finally, the authors would also like to thank Meta AI for the computing resources that made this work possible.

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

Table 3: Causal Discovery over $G$ for various pairs of attributes, $(m_1, m_2)$. For each pair, the top row corresponds to taking $m_1$ as the treatment whilst the bottom utilizes $m_2$ as the treatment. Please see appendix §F contains additional generated samples and a qualitative description of visual fidelity for each pair of attributes. Note, the results from this experiment are distinct from the same source or same destination experiments in 4.1 and summarizes the findings from answering **Q1-Q3** posed earlier.

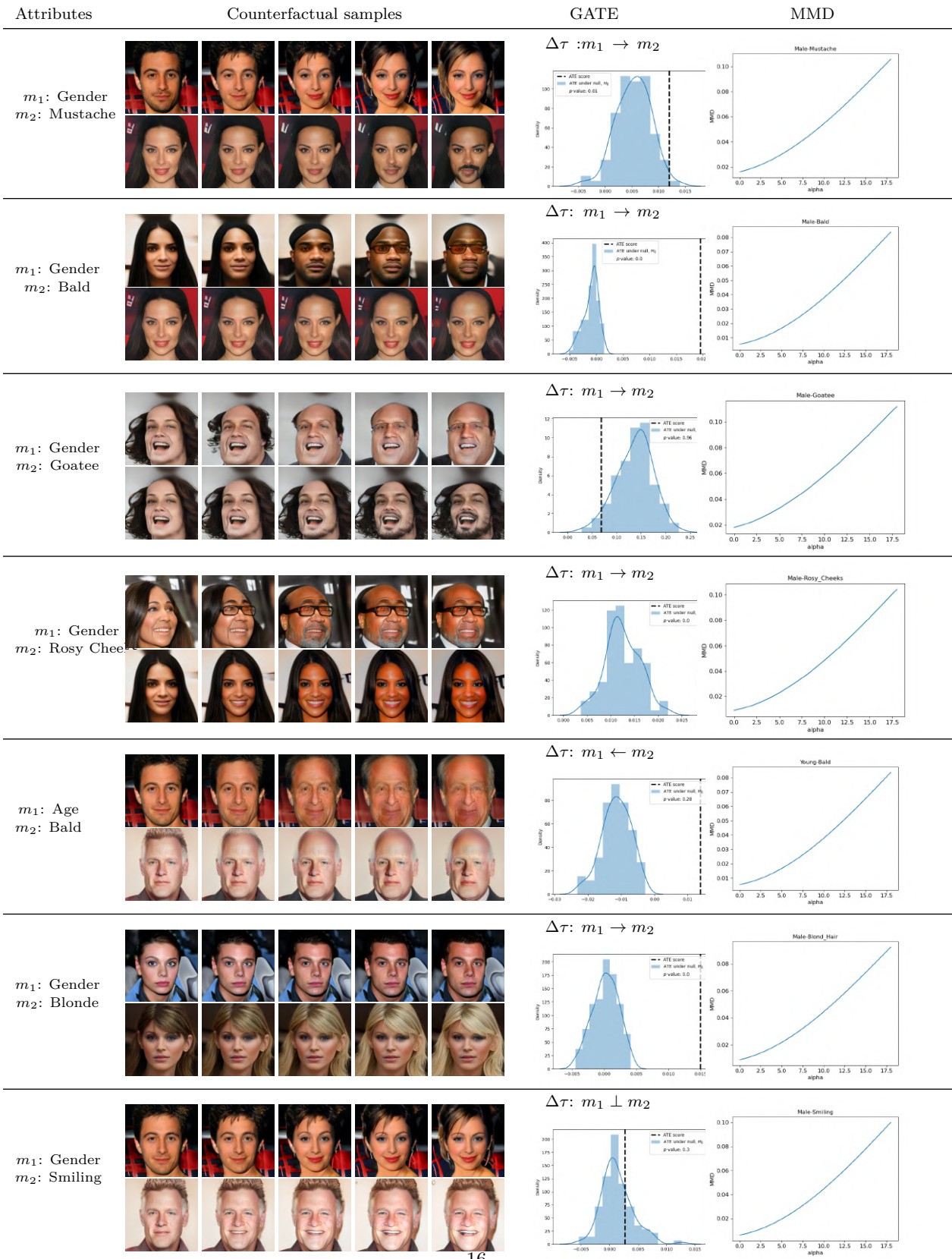

# A    Review: Casual inference and connections to generative models and CAGE

In this section we provide a brief overview of the causal inference methodology we rely upon for CAGE as well as provide intuitive explanations for our work. CAGE relies upon the framework of structural equation models (SEMs) which can be used to formalize causal knowledge as well as answer interventional and counterfactual queries. We provide a brief description below, for further details please see (Pearl, 2009) as well as (Pawlowski et al., 2020) and (Khemakhem et al., 2021) for applications in the context of deep generative models.

**Structural Equation Models**: For a $d$-dimensional random variable, $\mathbf{x} = (x_1, \ldots, x_d)$, a SEM is defined as a collection of structural equations:

$$\mathcal{S}_j : \quad x_j = f_j(\mathbf{pa}_j, n_j), \quad j = 1, \ldots, d$$

which dictate the generative mechanism for observed data, $\mathbf{x}$. We write $\mathbf{pa}_j$ to denote the causal parents of variable $x_j$ and introduce mutually independent latent noise variables (also referred to as latent disturbances) $n_j$ for each $x_j$ to accommodate non-deterministic causal relationships.

The set of structural equations, $\mathcal{S} = (\mathcal{S}_1, \ldots, \mathcal{S}_d)$, serves two purposes: first it encodes causal structure over observed random variables $\mathbf{x}$ following a given distribution $\mathbb{P}_\mathbf{x}$. As a concrete example, if $x_k \in \mathbf{pa}_j$ then $x_k$ is a direct cause of variable $x_j$. Second, it also defines a generative mechanism to sample new observations from the distribution $\mathbb{P}_\mathbf{x}$ by simply iterating through the set of structural equations, $\mathcal{S}$.

**Interventions and the do Operator**: One of the fundamental benefits of encoding causal structure via SEMs is that it we can directly parameterize interventions over the associated causal graph (Pearl, 2009). Formally, this involves re-defining the structural equation over the variable which we will intervene over. For example, if we considered the intervention of setting the $x_j = \alpha$, this would redefine $\mathcal{S}_j$ as follows:

$$\mathcal{S}_j : \quad x_j = \alpha.$$

Pearl (2009) denotes such a deterministic intervention as $do(x_j = \alpha)$.

**Counterfactuals**: Counterfactual queries seek to understand statements of the form: what would the value for variable $x_i$ have been if variable $x_j$ had taken value $\alpha$, given that we have observed $\mathbf{x} = \mathbf{x}_{obs}$? It therefore follows that counterfactual and interventional queries are closely related, with the fundamental difference being that an interventional query seeks to marginalize over latent variables, whereas a counterfactual query is evaluated conditional on latent variables inferred from observed $\mathbf{x} = \mathbf{x}_{obs}$. As such, a counterfactual query first infers latent disturbance variables, $n_i$, and then proceeds in the same manner as an intervention.

**Parameterization of Interventions and Counterfactuals in CAGE**: In the context of CAGE, we consider counterfactual queries over binary metadata variables, $m_c$. As described in Section 3, we first infer the latent associated latent disturbance variables by projecting a given image, $\mathbf{x}$, to a latent encoding $\mathbf{z} = \text{ENC}(\mathbf{x})$. Thereafter, it is possible to define an intervention over a given binary metadata attribute, $do(m_c = \alpha)$ in a variety of ways. In Section 3.3 we consider interventions parameterized by a linear projection over the latent variable, $\mathbf{z}$ such that $\mathbf{z}(\mathbf{do}(m_c = 1)) = \mathbf{z} + \alpha\mathbf{h}_c$ where the vector $\mathbf{h}_c$ is defined via training a linear classifier (see Figure 3).

# B    Extending Average Treatment Effects to Generative Models

For average treatment effects, we are interested in computing

$$\tau = \mathbb{E}_i \left[ Y_{m_c=1}(i) - Y_{m_c=0}(i) \right].$$

There are two important distinctions between the use of GATE in CAGE and a traditional ATE computation:

- First, as we are interrogating a generative model, we can observe *both* $Y_{m_c=1}(i)$ and $Y_{m_c=0}(i)$. This is a significant difference to standard ATE calculations, which can only observe one outcome.

- Second, we are interested in understanding implicit causal structure in the generator, $G$, the expectation over samples from the prior of $G$ as opposed to over samples from an observational dataset. This is an important distinction as it disambiguates the causal structure present in an observation dataset, which is the focus of traditional causal discovery methodology, with any causal structure implicit within the generator, which is our primary concern.

For unbiased estimation of ATE (a causal quantity) via statistical quantities, it is standard to assume unconfoundedness, positivity, and SUTVA. For the use of GATE in CAGE, we implicitly make further assumptions such as:

1. **Counterfactual parameterization**: The latent space of a generator, $G$, must be linearly separable with respect to the attributes of interest. This assumption is required as CAGE parameterizes the assignment of counterfactual treatments as linear manipulations over the latent space of a generator. We note that such an assumption has been shown to hold in practice for deep generative models (Denton et al., 2019; Ramaswamy et al., 2021).
2. **Complete support of the generator**: The generator, $G$, can generate high quality samples over the entire support of its latent space, $\mathcal{Z}$. This assumption is required to guarantee that linear manipulations over the latent space of a generator continue to produce high quality images, such that our resulting conclusions are not the result of e.g., image artefacts.
3. **Sufficiently expressive and accurate classifiers**: CAGE is premised on the use of probabilistic classifiers in order to quantify the presence/absence of an effect variable on counterfactual examples and thus compute treatment effects as specified in equations (6-7). We require accurate and calibrated classifiers (ideally, Bayes optimal) for computing GATE.

## C    Model Details

For our experiments, whenever possible, we used the default settings found in the original papers of all chosen models. In particular, we used the default settings for both Mintnet (Song et al., 2019) and StyleGAN2 (Karras et al., 2020) which were pretrained on MNIST and FlickrFacesHQ respectively. For Mintnet we finetuned on a MorphoMnist dataset for 250 epochs using the Adam optimizer with default settings. Similarly, we also finetuned StyleGAN2 on CelebAHQ for  2000 iterations using 10% of CelebAHQ. Our synthetic experiments on the other required us to train a Masked AutoRegressive Flow (Papamakarios et al., 2017) that consisted of 10 layers with 4 blocks per layer. The Masked AutoRegressive Flow was trained for 5000 iterations using 5000 data samples.

### C.1    Human Evaluation of Counterfactuals on Amazon Mechanical Turk

To study the visual quality of generated counterfactuals by manipulating the cause or effect variable as determined by CAGE we solicit human evaluators on the Amazon mechanical turk platform. In particular, we collect 100 generated pairs for each DAG in both the same source and same destination settings. Each pair of generated samples is then evaluated by 3 different human annotators giving a sum total of 300 annotations per DAG. A capture of the UI provided to the human annotators is depicted below 5.

**Instructions:** Given these pair of generated images, which one is more realistic when compared to actual human faces? Specifically, using your initial impressions and IGNORING THE BACKGROUND, which image is more likely to fool you as being a real person's picture if it were shown briefly. There is no right answer but please select the most realistic image using your own judgement.

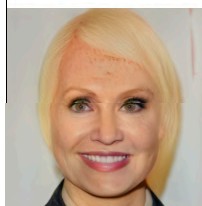

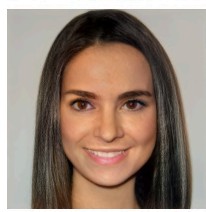

○ Top Image
○ Bottom Image

Submit

Figure 5: UI provided to AWS Turkers.

### C.2  $\Delta\tau$ **Mixture of Gaussians and MorphoMNIST**

To illustrate the fact that a deep generative model may learn a different causal relationship than the ground truth data generation process we consider the same mixture of gaussian experiment as the main paper. We consider both a causal ordering and anti-causal ordering over input variables that are fed into a Masked Autoregressive Flow model which is then trained using maximum likelihood. In each case CAGE reports $\Delta\tau > 0$ indicating for the same ordering of variables which indicates that it is possible for the MAF model to learn a different internal causal ordering than the ground truth.

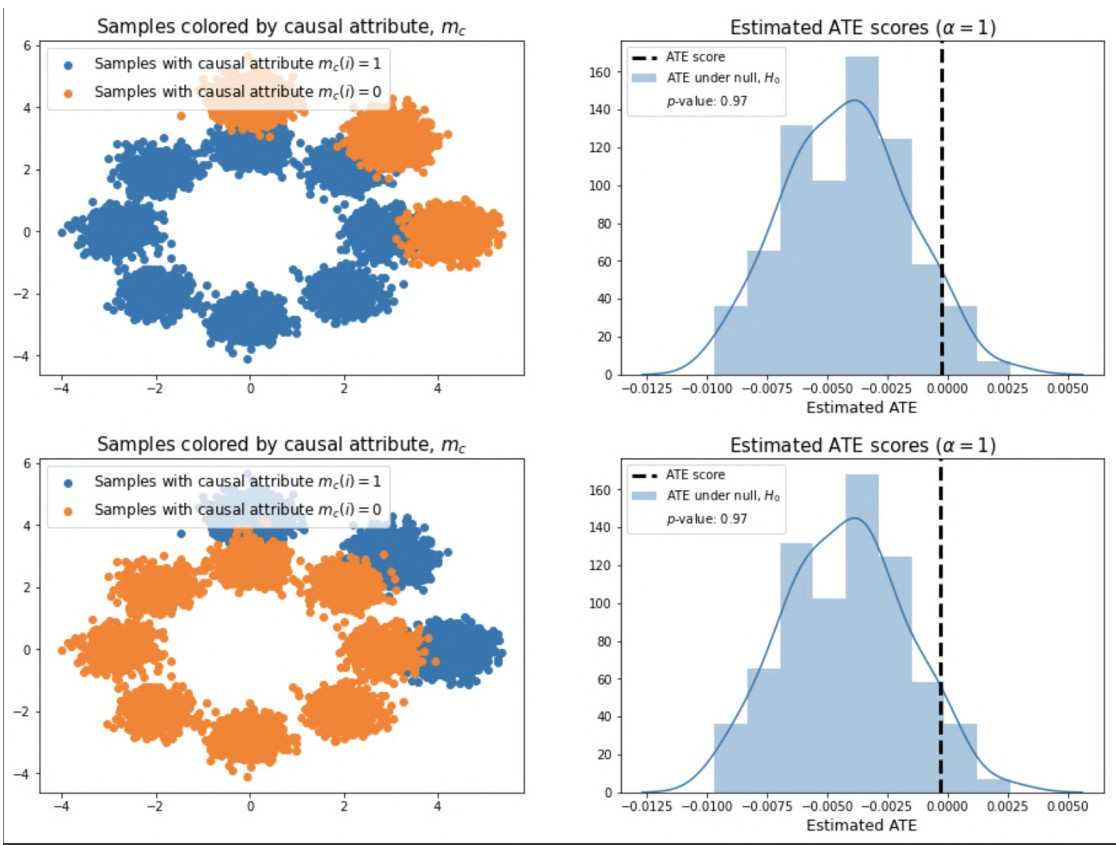

Figure 6: $\Delta\tau$ plots on Mixture of Gaussians for both causal and anti-causal input variable orderings for Mixed Autoregressive Flow generative model.

We plot the $\Delta\tau$ as a function of interpolant strength below for the MorphoMNIST dataset. As observed $\Delta\tau > 0$ which indicates that under our generator $G$ thickness is the cause of intensity which matches the true synthetic data generation process.

### **A note on baseline methods for CelebHQ**

We compared CAGE to well-established causal discovery algorithms when looking to infer structure over a deep generative model, $G$. As noted in §4, the causal discovery baselines considered operate over the class probabilities output by probabilistic classifiers trained as §3.4. We note that while it would be possible to learn causal structure using baseline methods, such as LiNGAM, over the metadata (e.g., hair color and gender), this would not necesarly provide any insights into the causal structure implicit within the generator, $G$. For this reason, we instead focus on applying causal discovery methods over the output of proxy classifiers.

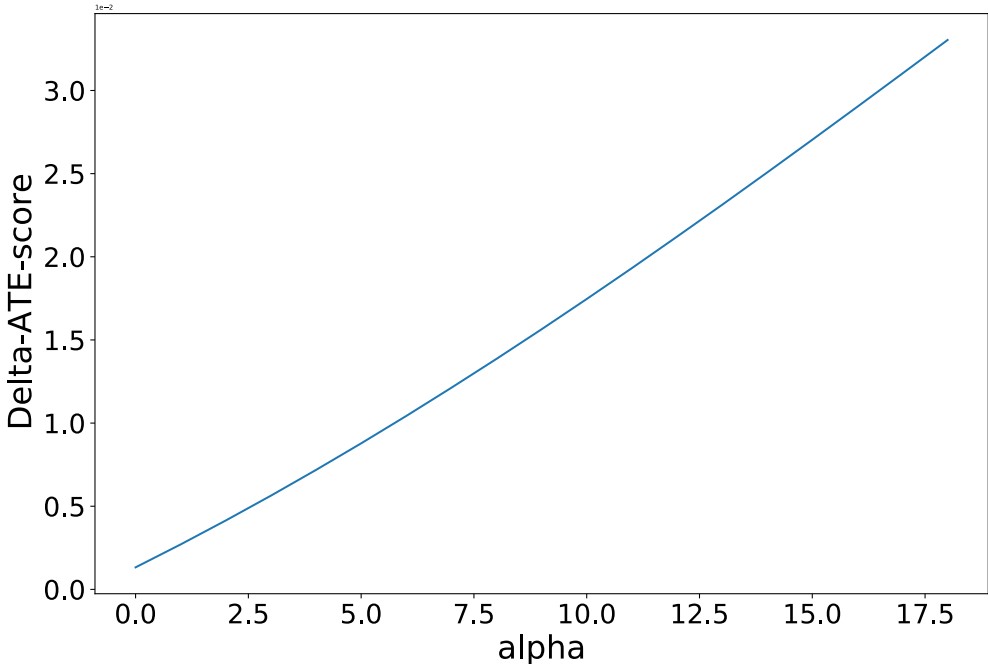

Figure 7: $\Delta\tau$ plot on the MorphoMNIST dataset as a function of $\alpha$.

## D Additional Baseline Experiments

In this section we perform causal discovery in the latent space using various popular approaches in the literature. As noted in the main paper these approaches cannot be applied to the observation space which are high dimensional images and are not baselines.

| Method | ANM | DAGs with No Tears | LinGAM |
|---|---|---|---|
| Gender, Mustache | $\longrightarrow$ | $\longrightarrow$ | $\longrightarrow$ |
| Gender, Bald | $\longrightarrow$ | $\longrightarrow$ | $\perp$ |
| Gender, Goatee | $\longleftarrow$ | $\longrightarrow$ | $\longleftarrow$ |
| Gender, Rosy Cheeks | $\longleftarrow$ | $\longrightarrow$ | $\longleftarrow$ |
| Age, Bald | $\perp$ | $\longrightarrow$ | $\perp$ |
| Pointy Nose, Brown Hair | $\longrightarrow$ | $\longrightarrow$ | $\perp$ |
| Gender, Smiling | $\longrightarrow$ | $\longrightarrow$ | $\perp$ |

**Hard vs. Soft**. As we have access to ground truth labels it is tempting to consider whether using hard labels as opposed to the classifiers probability is better suited to computing our $\Delta\tau$ metric. In the table below we repeat our causal discovery experiment over CelebAHQ. As observed, all baselines provide unreliable estimates to the causal relationship between variables when compared to the main table which is computed using soft labels. Finally, we found it useful to assign soft labels provided by the classifiers used in the ATE computations for all augmented images during group DRO training.

**Baselines without using CounterFactuals**. We now turn to the use of Counterfactuals when computing our baseline scores. Specifically, we attempt causal discovery purely using observation data and labels with both hard and soft labels, which is in contrast to the main paper which considered baselines which had access to classifier probabilities on counterfactual data. The table below shows this result of this ablation.

| Method | ANM | DAGs with No Tears | LinGAM |
|---|---|---|---|
| Gender, Mustache | ⊥ | ⟶ | ⊥ |
| Gender, Bald | ⊥ | ⊥ | ⊥ |
| Gender, Goatee | ⊥ | ⊥ | ⊥ |
| Gender, Rosy Cheeks | ⊥ | ⟵ | ⊥ |
| Age, Bald | ⊥ | ⟵ | ⟶ |
| Pointy Nose, Brown Hair | ⊥ | ⟵ | ⟶ |
| Gender, Smiling | ⊥ | ⊥ | ⊥ |

| Method | ANM | DAGs with No Tears | LinGAM |
|---|---|---|---|
| Gender, Mustache | ⟶ | ⟶ | ⊥ |
| Gender, Bald | ⟶ | ⟶ | ⊥ |
| Gender, Goatee | ⟵ | ⟶ | ⊥ |
| Gender, Rosy Cheeks | ⟵ | ⟶ | ⊥ |
| Age, Bald | ⊥ | ⟶ | ⊥ |
| Pointy Nose, Brown Hair | ⊥ | ⟵ | ⊥ |
| Gender, Smiling | ⟶ | ⟶ | ⟶ |

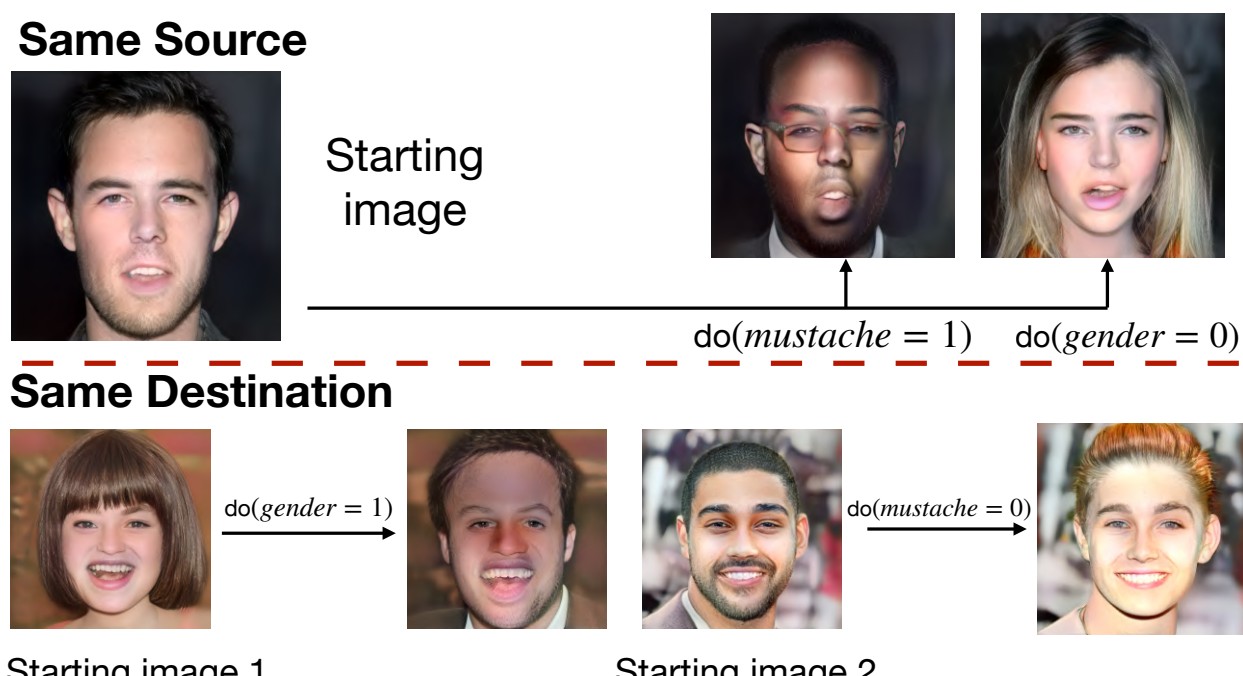

Figure 8: A visual description of the same source and same destionation settings. In the same source setting we start of with the same image which is then intervened using each meta-attribute separately. In teh same destination setting we start out with two images which are then intervened in such a way that the output of both interventions arrives at the same combination of meta-attributes. For example, we may wish to generate samples of men without mustaches by intervening on a starting image of a female and then changing gender or starting with an image of a male with a mustache and intervening on the mustache meta-attribute.

## E Qualitative Samples for Same Source and Same Destination

In this section we provide selected qualitative examples that were used for our human evaluation study on Amazon Mechanical Turk. In each visualization we show a panel of three images; the leftmost image is the source image while the middle and rightmost image correspond to manipulating the first and second meta-attribute respectively. We also give a visual description of both the same source and same destination setting below:

### E.1 Same Source

**Gender→ Mustache**.

**Gender→ Goatee**.

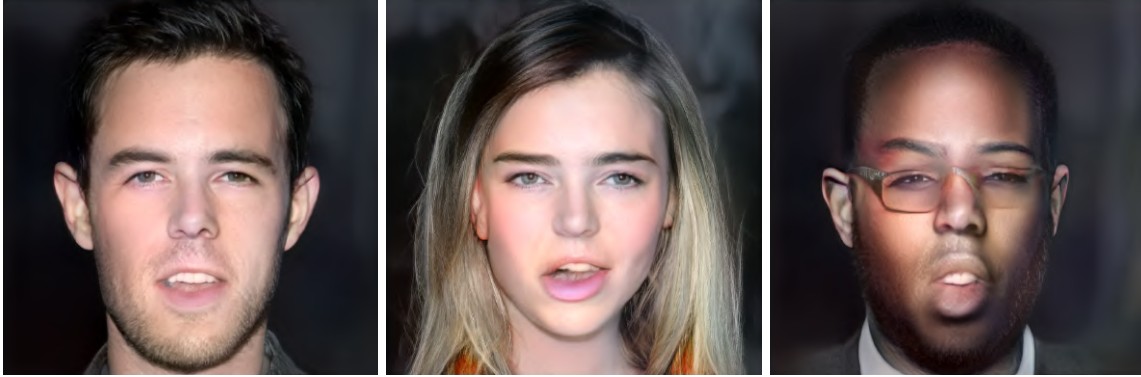

Figure 9: **Left:** The target image used for source manipulation. **Mid:** Manipulating the causal Gender variable. **Right:** Manipulating the effect Mustache variable.

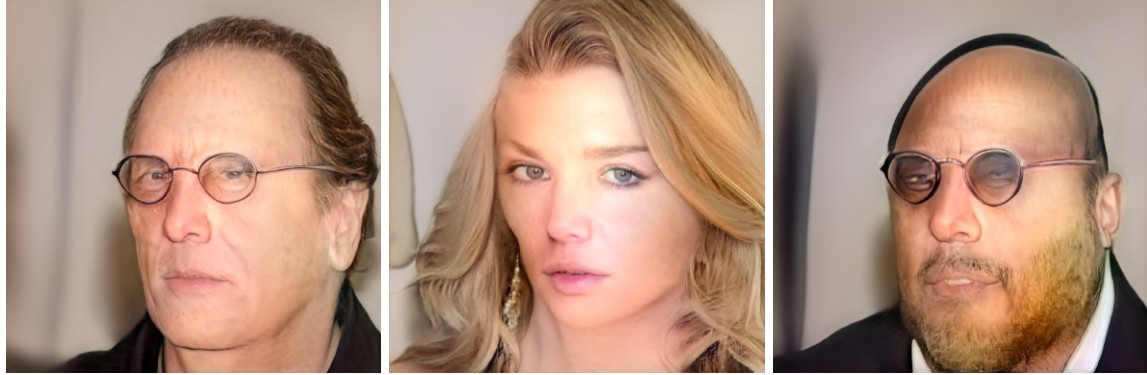

Figure 10: **Left:** The target image used for source manipulation. **Mid:** Manipulating the causal Gender variable. **Right:** Manipulating the effect Goatee variable.

**Gender→ Bald**.

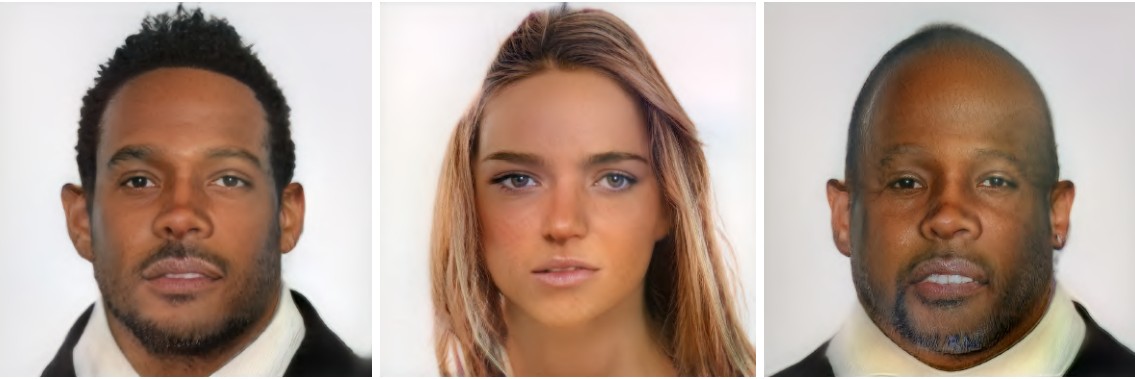

Figure 11: **Left:** The target image used for source manipulation. **Mid:** Manipulating the causal Gender variable. **Right:** Manipulating the effect Bald variable.

**Gender→ Blonde Hair**.

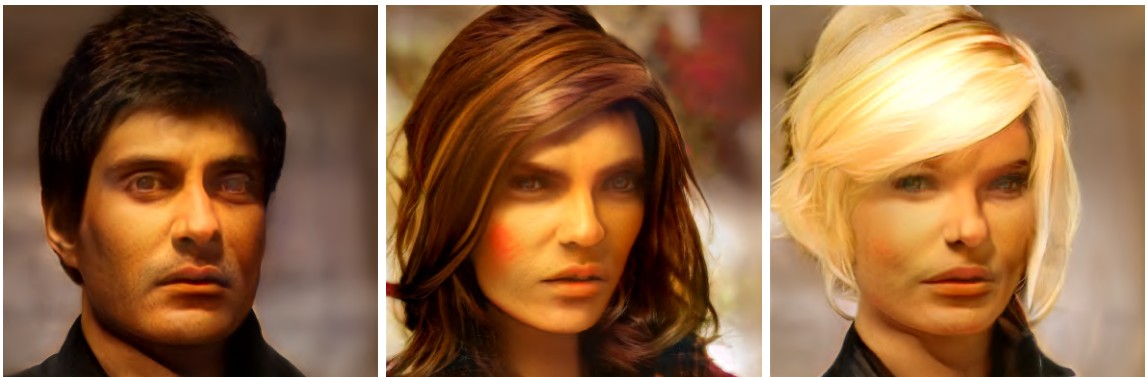

Figure 12: **Left:** The target image used for source manipulation. **Mid:** Manipulating the causal Gender variable. **Right:** Manipulating the effect Blonde Hair variable.

**Gender→ Rosy Cheeks**.

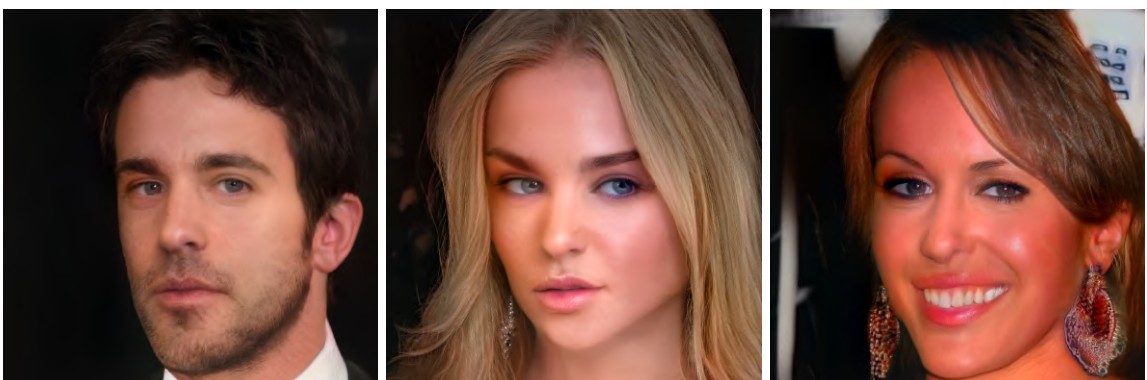

Figure 13: **Left:** The target image used for source manipulation. **Mid:** Manipulating the causal Gender variable. **Right:** Manipulating the effect Rosy Cheeks variable.

**Age← Bald**.

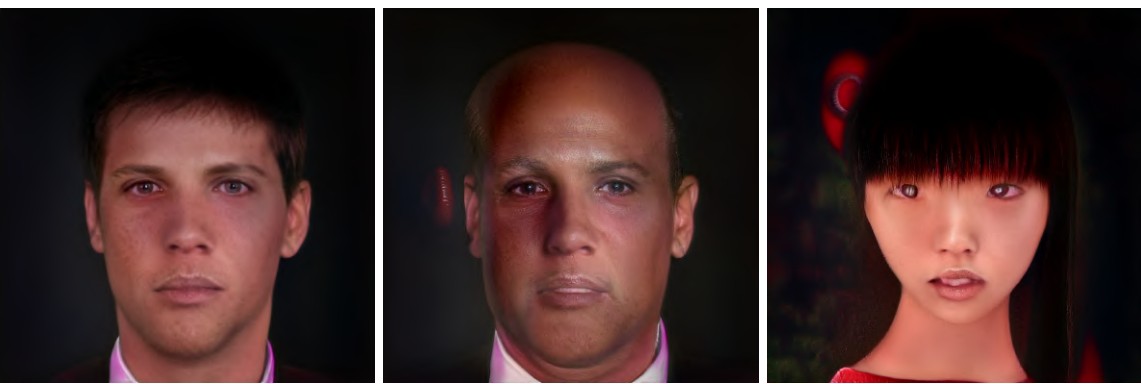

Figure 14: **Left:** The target image used for source manipulation. **Mid:** Manipulating the causal Bald variable. **Right:** Manipulating the effect Age variable.

### E.2 Same Destination

**Gender→ Mustache**.

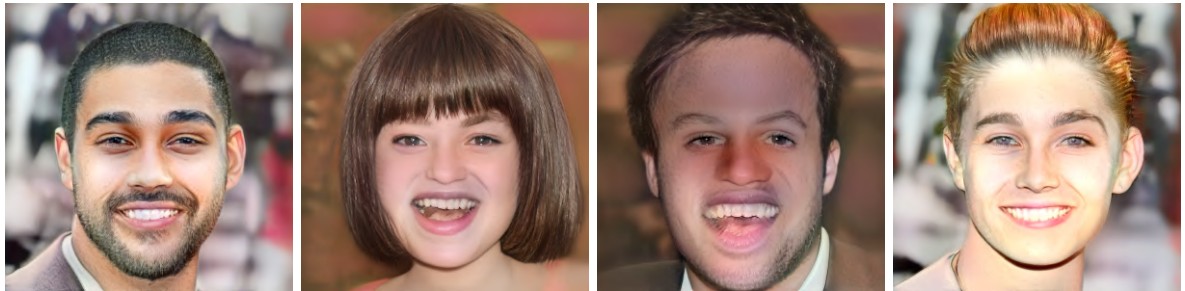

Figure 15: **Left and Mid Left:** The target images used for same destination manipulation. **Mid Right:** Manipulating the causal Gender variable. **Right:** Manipulating the effect Mustache variable.

**Gender→ Goatee**.

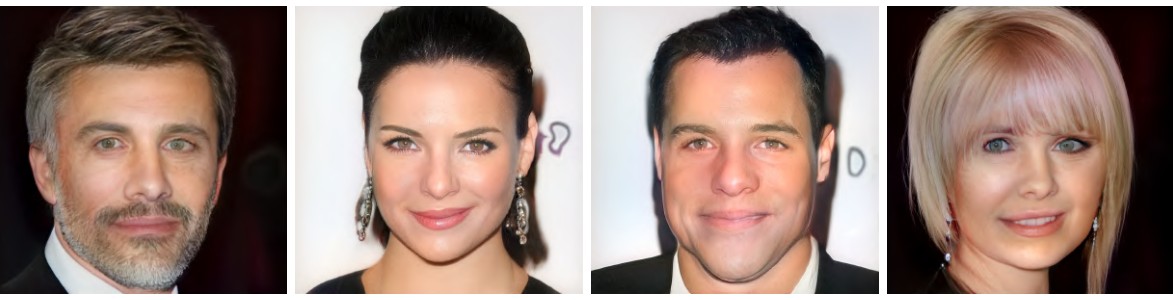

Figure 16: **Left and Mid Left:** The target images used for same destination manipulation. **Mid Right:** Manipulating the causal Gender variable. **Right:** Manipulating the effect Goatee variable.

**Gender→ Bald**.

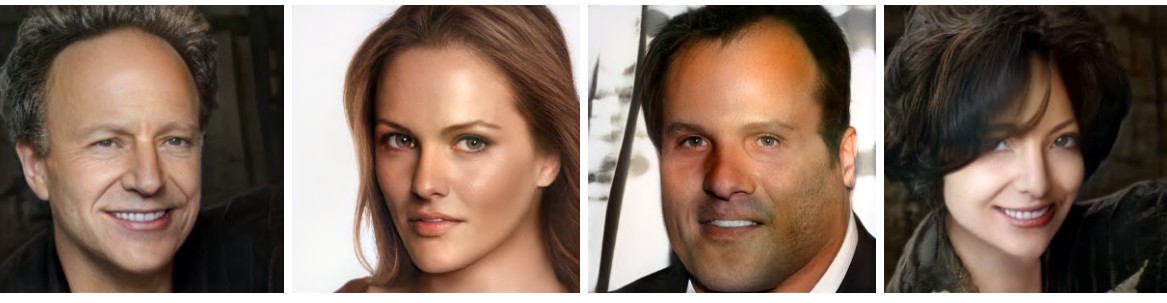

Figure 17: **Left and Mid Left:** The target images used for same destination manipulation. **Mid Right:** Manipulating the causal Gender variable. **Right:** Manipulating the effect Bald variable.

**Gender→ Blonde Hair**.

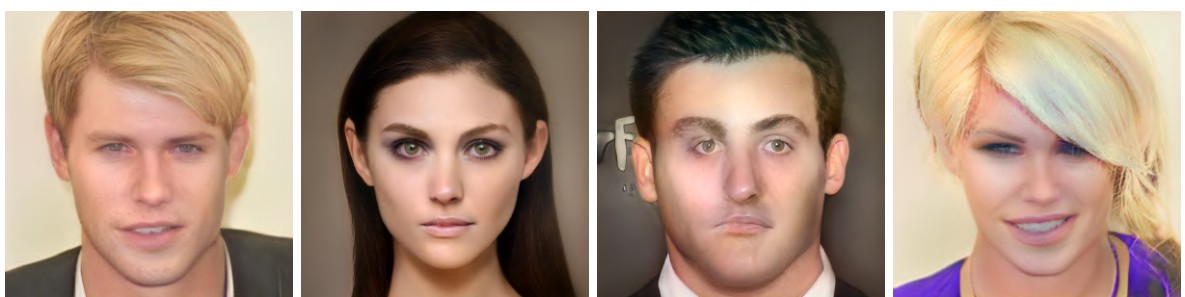

Figure 18: **Left and Mid Left:** The target images used for same destination manipulation. **Mid Right:** Manipulating the causal Gender variable. **Right:** Manipulating the effect Blonde Hair variable.

**Gender→ Rosy Cheeks**.

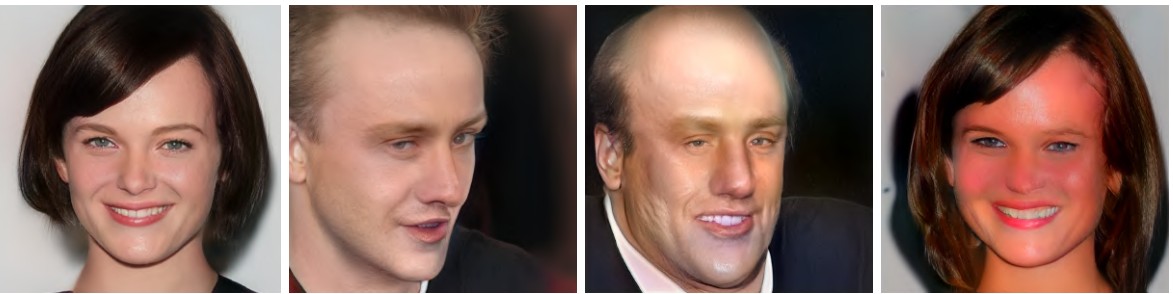

Figure 19: **Left and Mid Left:** The target images used for same destination manipulation. **Mid Right:** Manipulating the causal Gender variable. **Right:** Manipulating the effect Rosy Cheeks variable.

**Age ← Bald**.

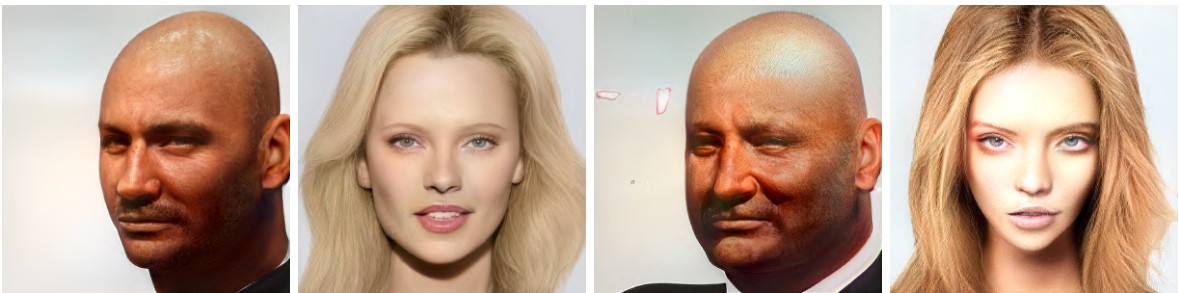

Figure 20: **Left and Mid Left:** The target images used for same destination manipulation. **Mid Right:** Manipulating the causal Bald variable. **Right:** Manipulating the effect Age variable.

## F  Additional Generated samples

We now provide additional qualitiative examples of the generated counterfactuals on the same set of DAG's as considered in the main text. In the first row of all settings we generate counterfactuals by manipulating the first attribute variable (e.g. Gender) while the second row corresponds to counterfactuals where the second attribute is manipulated. The third row can either be the first or second attribute depending on context. We note that the generated samples may expose unintended biases that may be present in CelebAHQ and can be learned by the generator. While CAGE does not introduce any new sources of biases it should be noted that generating counterfactuals in this manner can further exacerbate these initially hidden biases. As a result, CAGE can serve as an important aid in empirically illuminating important ethical concerns that should be taken into account for any purposeful uses of generative models.

**Gender, Mustache**.

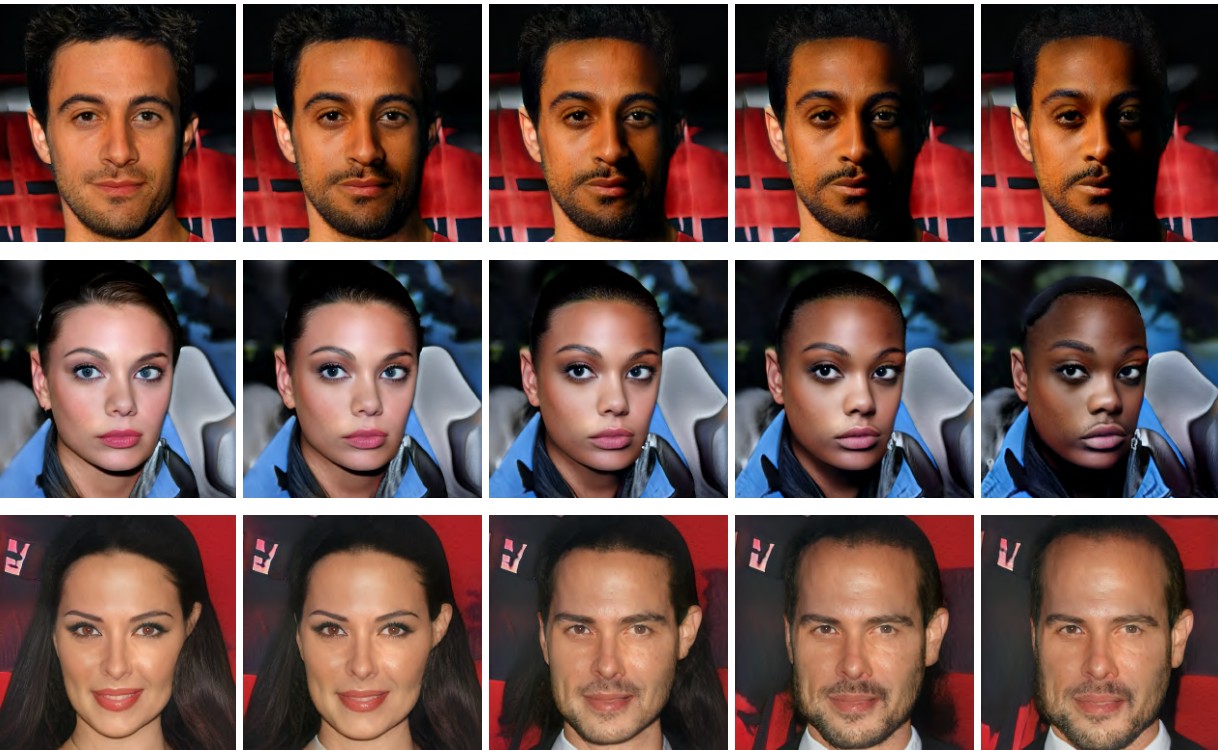

We consider generating samples by either manipulating the mustache attribute (top two rows) and then manipulating gender. As we can see, when the Gender is male we are able to semantically preserve the gender as we increase the presence of mustache (top row). Similarly, we find that when the gender is female we are still able to semantically preserve the (subjective) gender as we manipulate the mustache attribute (middle row). Finally, in the third row we change the gender of from female to male and notice that although the mustache attribute was not explicitly manipulated it appears as we manipulate gender more aggressively to male (bottom row). These results suggests that the generative models learned causal relationship is that *gender causes mustache*.

**Gender, Goatee**.

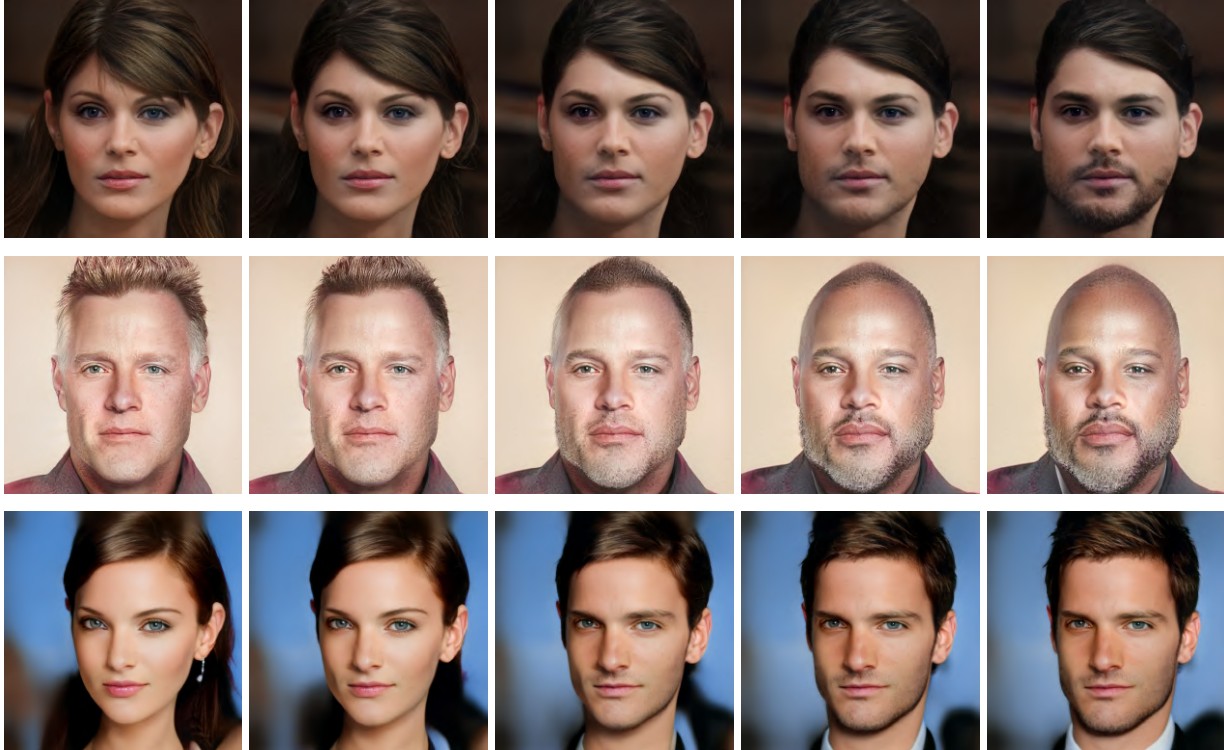

We consider generating samples by either manipulating the goatee attribute (top two rows) and manipulating gender. In the first row we see that manipulating the goatee attribute leads to a visual and subjective change the samples perceived gender (top row). Similarly, we find that when the gender is already male increasing the presence of a goatee does not reverse the gender to female as expected (middle row). Finally, in the third row we change the gender of from female to male and notice that there is no increase in goatee in the generated samples. These results suggests that the generative models learned causal relationship is that *gender is the cause goatee.*

**Gender, Smiling**.

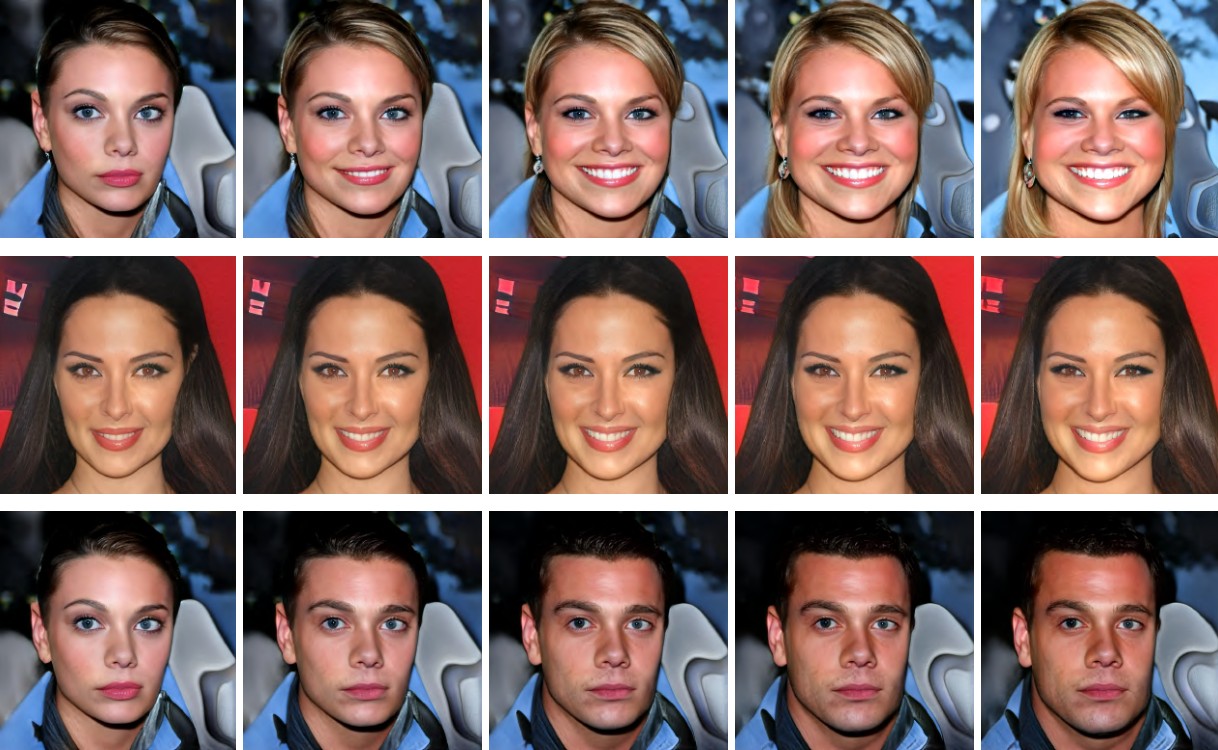

We consider generating samples by either manipulating the smiling attribute (top two rows) and manipulating gender. In the first two rows we see that increasing the smiling attribute does not cause a (subjective) change the samples perceived gender. Conversely, in the last row manipulating a samples gender from female to male preserves the smiling attribute (or lack thereof) in the generated samples. These results suggests that the generative models learned causal relationship is that *gender is independent of smiling*.

**Age, Bald**.

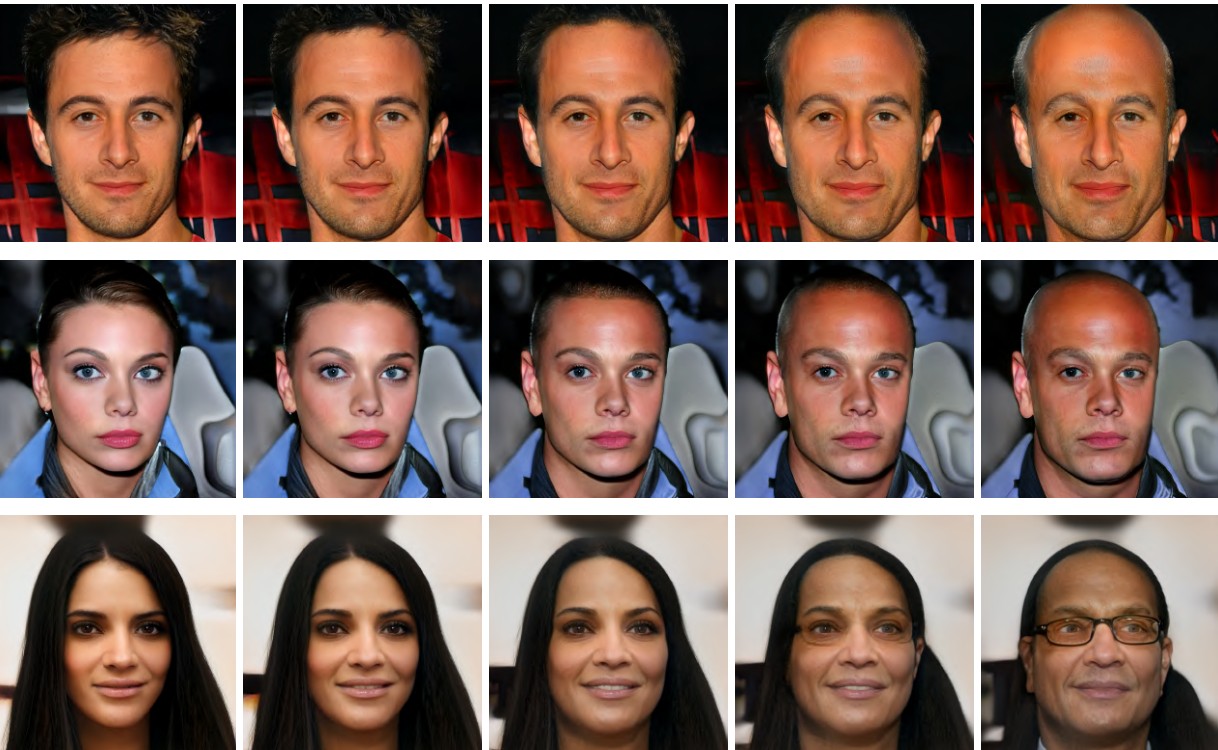

We consider generating samples by either manipulating the baldness attribute (top two rows) and then manipulating age (last row). In the first two row we see that increasing the baldness attribute does considerably change the age of the generated sample. On the other hand, in the last row we increase the age of the candidate and observe that while the baldness attribute increases it is not the extent as samples found in the first two rows. These results suggests that the generative models learned causal relationship, in contrast to our biological prior, is that *baldness is the cause of age*.

**Pointy Nose, Brown Hair**.

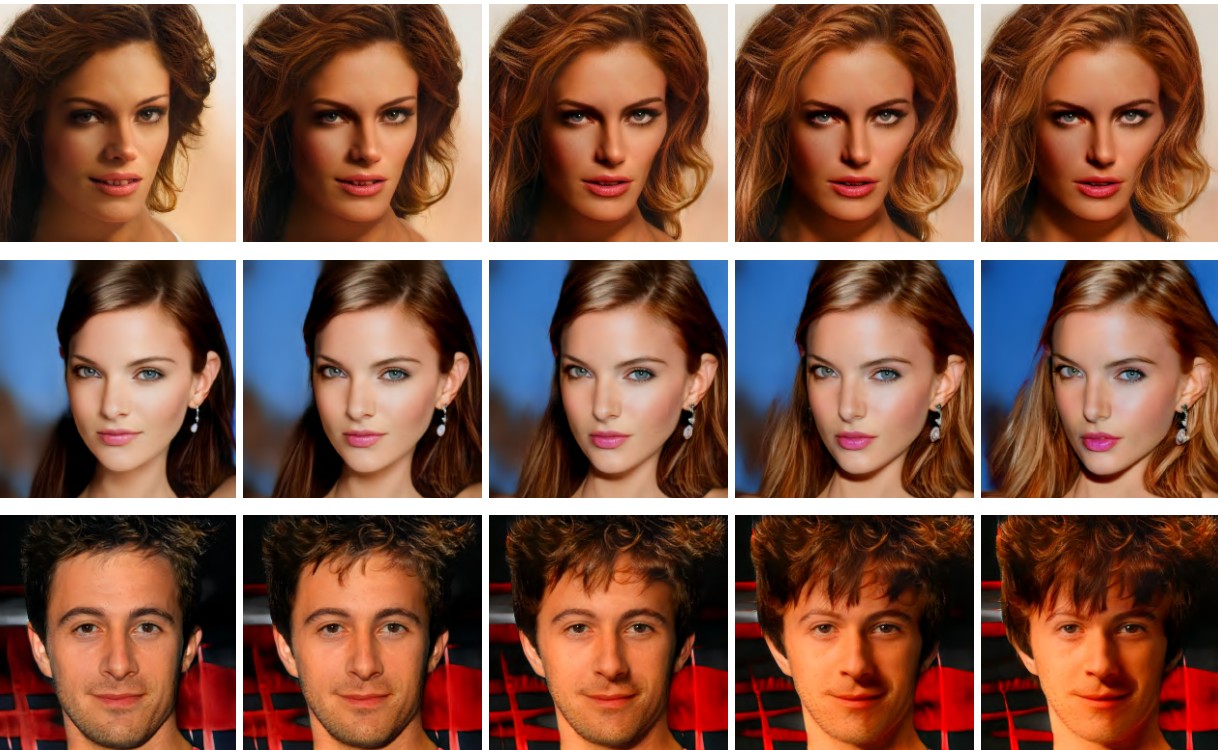

We consider generating samples by either manipulating the pointy nose attribute (top two rows) and manipulating brown hair. In the first two rows we see that increasing the pointy nose attribute does cause a change in the samples hair color from dark brown to more blonde. Conversely, in the last row manipulating a samples hair color to be more brown does not appear to significantly change the pointy nose attribute in the generated samples. These results suggests that the generative models learned causal relationship is that *pointy nose causes brown hair*.

**Gender, Rosy Cheeks**.

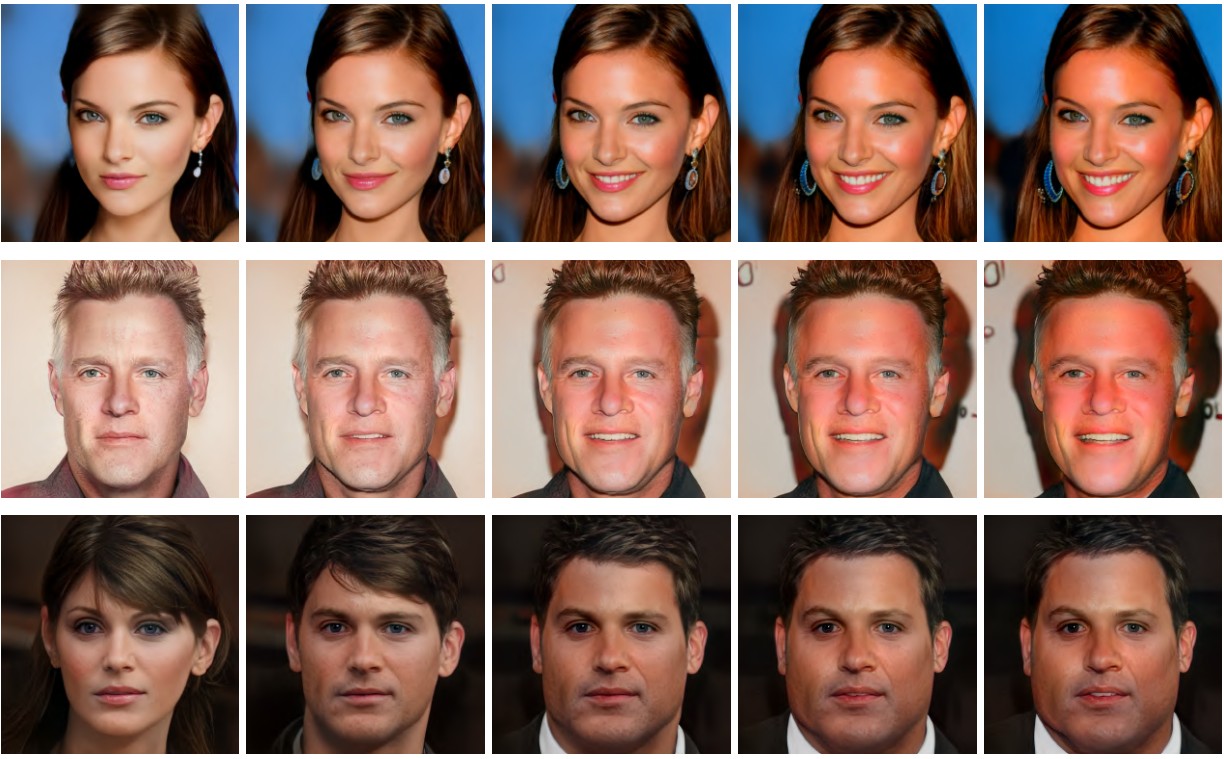

We consider generating samples by either manipulating the rosy cheek attribute (top two rows) and manipulating gender. In the first two rows we see that increasing the rosy cheek attribute does not cause a change in the samples (subjective) gender—i.e. both females and males retain their gender as we increase their rosy cheeks. On the other hand, in the last row manipulating a samples gender from female to male appears to slighltly decrease the rosy cheek attribute in the generated samples. These results suggests that the generative models learned causal relationship is that *gender causes rosy cheeks*.

**Gender, Blonde Hair**.

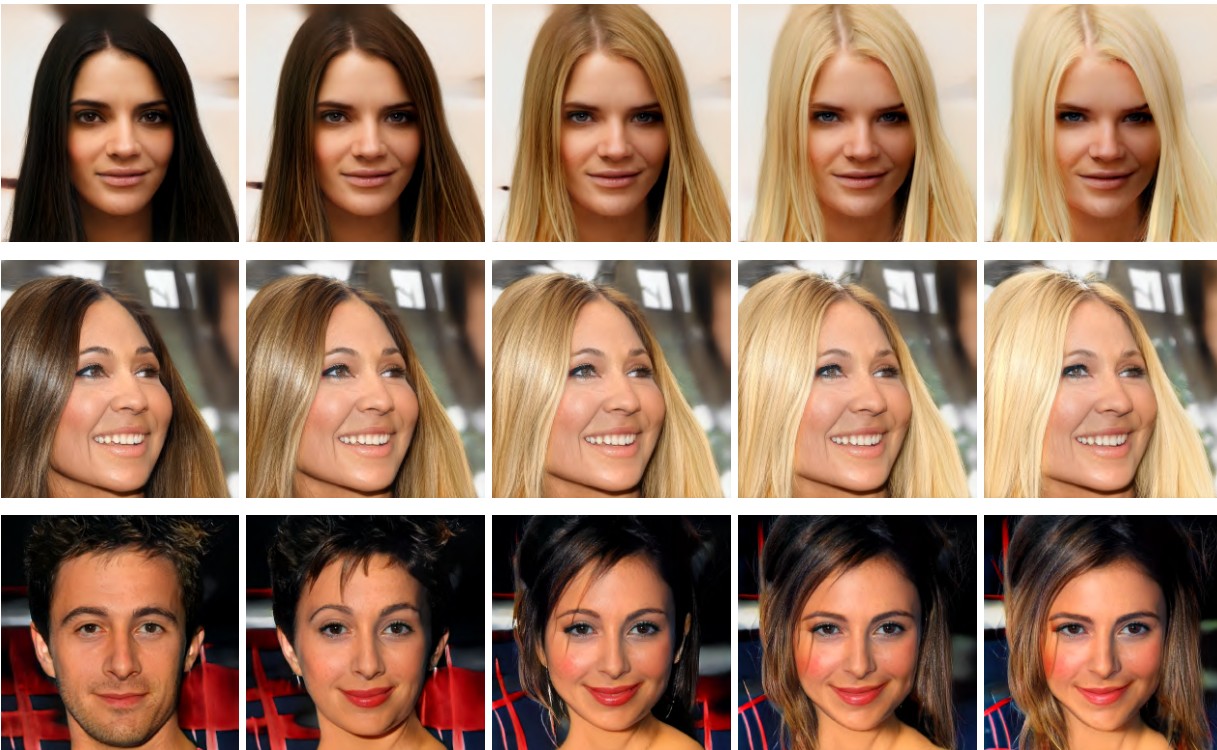

We consider generating samples by either manipulating the blonde hair attribute (top two rows) and manipulating gender. As observed in the first two rows increasing the blondeness of a samples hair does not change the perceived gender from female to male. In sharp contrast, the last row we see that manipulating the gender from male to female leads to the samples hair color changing towards being more blonde. These results suggests that the generative models learned causal relationship is that *gender causes blonde hair*.

## Ethical Implications of CAGE

As deep generative models grow in their practicality and deployment in high-stake applications, it is important to take a critical view of the inherent biases implicitly encoded in these models (Sheng et al., 2019; Choi et al., 2020; Bender et al., 2021). Our framework allows for a principled study of such biases through a causal analysis, potentially paving the way to mitigate the undesirable harms of deep generative models.

For example, our approach highlights important failings of the generative model itself—e.g. when cause-effect relationships are inconsistent with biological priors for human subjects or with physical relationships for scientific applications. In such cases, the use of these models in any high-stakes downstream application can be very costly without any intervention. Examples range from the current use of generative models for accelerating scientific discovery as well as societal applications in training fair classifiers. Ironically, in all such applications, the use of generative models can worsen the problem at hand by introducing new blindspots through the use of generated data that does not conform to the desired cause-effect relationship. Explicitly inferring these failings provides researchers and practitioners with an opportunity to develop algorithms and evaluation metrics that go beyond the current focus on visual fidelity or likelihoods as well as derive better data acquisition strategies for finetuning generative models.

On the other hand, we acknowledge that like many other efforts in generative modeling, our endeavors for better understanding these models can potentially be exploited by malicious actors for generating fake content, commonly known as Deepfakes (Korshunov & Marcel, 2018). Finally, causality has proven to be a highly powerful tool for formalizing important concerns of machine learning models with regards to fairness, robustness, and interpretability. However, as highlighted in our limitations, any causal framework including ours itself implicitly encodes some assumptions that may not hold in practice. Hence, researchers and practitioners should exercise due caution and appropriate judgment in interpreting the findings.

We would also like to highlight our proposed framework and its application to natural images has been constructed for illustrating this research and not for any business application.

