# OpenReview forum: "Controllable Generative Modeling via Causal Reasoning"
_TMLR — Accepted by TMLR_

### Review · Reviewer_inuw · 2022-10-19

**Summary Of Contributions:**

The goal of this paper is to use ideas from the causality literature to make pretrained latent variable models like GANs and VAEs more interpretable and to make their generation process controllable. The paper focuses on causal relations between a predetermined set of binary attributes such as gender or hair color that are provided alongside the main data (e.g. images of faces). Causal effects among such attributes can be defined by giving an intervention operation in the latent space. In practice this means taking the latent vector z representing an image, adding a vector h_c which is the normal of a classifier hyperplane (e.g. moving in the "blonde" direction of latent space), and then measuring the effect on the attributes using a latent-space classifier. This is a unique setting, because normally we can only execute one intervention per unit (image, person, ...) whereas here we can apply any number. Using this notion of intervention, a new controllable generation method is presented. Using a study with human raters, it is shown that intervening on the cause variable generally leads to better samples.

**Broader Impact Concerns:**

No concerns

**Requested Changes:**

Minor issues:
- Top of page 5, the union operation creates a big set with lots of vectors and metadata but no pairing. Better to use a set of tuples or something.
- Page 6, typo: "factual treats"

**Strengths And Weaknesses:**

The paper addresses an interesting question, namely how generative models can be made more interpretable and controllable using methods from causal inference. I did not find any significant technical errors, and the paper is mostly well written. Given the interest in causality and generative modelling I'm confident some readers would be interested in this paper.

The notion of intervention and causality employed here is precisely defined, in fact more precisely than in the wider causality literature. This is because in this setting the intervention is actually defined as a transformation on latent space (adding a vector h_c), whereas typically it is only operationally defined (e.g. give a patient a treatment). However it is not so clear that the result is in line with our intuition. For example, what does it mean to say that gender causes hair color (section 3)? Isn't every combination equally well possible? And couldn't we imagine intervening on one without changing the value of the other? So as the authors mention, the notion of causality as employed in this paper really only says something about the model, and how the operation of adding h_c changes the causal variables, rather than something about causality in the real world. As such one could argue about how important or interesting the notion of causality used in this paper is, but that would lead to a rather subjective discussion.

The main experiment (4.1) shows that according to human raters, sample quality is generally better when applying an intervention on a value deemed cause rather than on one deemed effect (in the same source setting). This seems interesting but I wonder what the explanation for this is? Could it be related to something simple like different frequency of occurence in the dataset (e.g. few to no women with mustaches)?

---

> ### Author Response · Authors · 2022-10-31
> **Rebuttal**
>
> We thank the reviewer for their positive appraisal of our paper! We are especially thrilled that the reviewer felt that our paper addresses an interesting question that some members of the community would find interesting. We now address the key clarification points.
>
> **”It is not so clear the result is in line with our intuition”**
>
> As the reviewer correctly points out, the notion of causality in our work is only with respect to the model and not the real causal mechanism. For instance, this information could aid us in identifying unintended biases encoded within our generative model which don’t respect real-world cause and effect relationships. As for intervening on every possible attribute while this is certainly possible in our setup the main insight is that intervening on a parent (causal variable) leaves a measurable signature on the child (effect variable) while the reverse is not true.
>
> **Intervening on the causal variable leads to better samples in the same source setting**
>
> The reviewer correctly points out one of our key results that modifying the causal variable in the same source setting leads to better visual fidelity as rated by human evaluators. Furthermore, it is interesting to consider to what extent the frequency of meta-attributes contribute the causal relationships found by CAGE. Indeed, as deep generative models are trained using statistical objectives the degree of an imbalance in a meta-attribute may contribute to the results but it does not necessarily imply that this signal is an indicator of cause effect. Case in point, if we consider attributes age and baldness (which are more balanced than gender and moustache), then a priori we may the causal association (if any) to be that age causes baldness. However, CAGE reports that in fact the implicit causal structure within a deep generative model is the opposite: $\text{age} \leftarrow \text{baldness}$. We further exploit this insight to perform controllable generate and produce a series of samples that humans rate as being higher quality.
>
>
> **Typos**
>
> We thank the reviewer for catching this. We have updated the draft to fix this.

---

### Review · Reviewer_FwcQ · 2022-10-21

**Summary Of Contributions:**

The paper propose CAGE, a framework for CAusal probing of deep GEnerative models. Compared with traditional causal discovery and causality, CAGE inferred implicit causal relationships in deep generative models and was able to discover the implicit causal structure of any generative model. The authors mainly consider two scenarios of control generation, source manipulation, and destination manipulation. The knowledge of causal directions inferred by CAGE can significantly improve the generation quality on the CelebAHQ dataset. Furthermore, the improvement of the source and destination manipulation scenarios was measured by extensive human evaluation. In addition, the authors extended the potential outcome framework to fit deep latent variable generative models. By computing the difference in treatment effects between the generative average treatment effect (GATE) of two variables to measure the candidate causal attributes.

Although the topic is interesting, the motivation and the intuition of the work are rather confusing. I pose a negative attitude towards the manuscript unless the authors can address my concerns.


**Broader Impact Concerns:**

I see no broader impact concerns.

**Requested Changes:**

1. Highlight the contribution and clearly present intuition.
2. Add a short overview of the experiments at the beginning of section 4.
3. Add a clearer description of Table 3.
4. Add clear explanations to things that appear for the first time.
5. Pay attention to some format uniformity.
6. Need to clarify some options in the paper.


**Strengths And Weaknesses:**

Strength
1. I agree with the author's opinion that traditional causal discovery techniques can damage generalization ability.
2. There are rich experimentation and clear visualization.
3. Appendix E is well explained.

Weakness
1. The motivation is confusing. The reasonability behind the assumption “we do not expect that the generative model perceives these meta-attributes as independent” needs to be clarified. Intuitively, meta-attributes are independent, (e.g., gender and hair color are unrelated), so meta-attributes have no cause-effect relationships.
2. The intuition of the work should be clearly presented. The authors claim that using causal insights can obtain fine-grained generation. However, by reading the Introduction part, it is totally unclear why and how can implicit causal relationships achieve the goal.
3. The merits of the proposed one are unclear. In the authors’ words, implicit causal structure within a deep generative model cannot well respect the true causal structure, so why do the authors still try to infer latent causal structure within a deep generative model?
4. It is necessary to explain these datasets and experiments at the beginning of section 4 so that the reader has a clearer sense of the functions of these experiments.
5. In Section 4.1, Experiment Design, there are two experimental setups: same source and same destination manipulation. In Table 3, it is unclear which one refers to the same source or destination.
6. The caption 'The target image used of source manipulation' in the figures shown in Appendix D is not mentioned and explained earlier, causing some confusion.
7. Given that the other places are written as 'Figure x.' it is suggested to change the last sentence of subsection 3.3 from 'Fig. 3' to 'Figure 3'.
8. Two candidate options that act directly on the counterfactual latent are proposed in subsection 3.4. I am not quite sure which option is adopted in the calculation.

---

> ### Author Response · Authors · 2022-10-31
> **Rebuttal**
>
> We thank the reviewer for their detailed comments and helpful feedback on the draft of our paper. We appreciate the reviewer’s opinion regarding the difficulty in applying traditional causal discovery methods to modern generative models. Moreover, we thank the reviewer for acknowledging the “richness of our experiments and their visualizations”. We now turn to the main questions raised by the reviewer:
>
> ***Conceptual Clarifications***
>
> **Q1. Motivation is unclear**
>
> As evident, today’s generative models have matured to an extent that they are being used in several downstream applications for data generation. Their direct use calls into question what are the properties of generated datasets and whether/if they differ from our expectations. We agree attributes such as gender and hair color could be independent in the real world, however as far the default training of many standard generative models in use today does not explicitly ensure this is the case for the generated data. This is precisely the motivation for our work, to reason about the cause-effect structure (or the lack of it as evidenced by an insignificant permutation test) that holds within a black-box pretrained latent variable generative model. We would like to highlight here many reasons for generative models to learn spurious causal structures e.g., dataset imbalances that are widely known in CelebA; imperfections in training algorithms such as mode collapse, etc.
>
>
> **Q2. Intuition is unclear**
>
> We recognize the reviewer's concern that the intuition for how the knowledge of cause-effect relationships can inform controllable generation might have been initially unclear from our introduction. As we demonstrate in our experiments in section 4.1, the knowledge of cause and effect relationships as learned by a generative model can be leveraged within a specific downstream application. For example, if the goal is to use the generative model for data augmentation—i.e. the same source setting—our human study informs us that intervening on the causal variables can often lead to samples with higher visual fidelity. Meanwhile, if the downstream task is in-painting or only modifying minor details without affecting the image globally it is easier to intervene on the effect variable which leads to more fine-grain control over the generated samples.
>
> **Q3. Implicit causal structure within a deep generative model cannot well respect the true causal structure**
>
> This is related to the first question posed by the reviewer. We recognize the reviewers' concern with respect to the merits of our proposal but we believe there is a potential source of confusion. Our CAGE framework seeks to infer $\textit{implicit}$ causal associations within a pre-trained generative model which may not respect the ground truth causal structure. We argue that this is still a fundamental problem that can a) be used to identify unexpected biases within a deep generative model and b) obtain better samples as we demonstrate via controllable generation. The thesis of this work serves to demonstrate that knowledge of cause-effect relationships can also be tied to a pretrained model while past work is tied to the actual data. Moreover, extracting the model’s causal relationship benefits the practitioner by informing them when the generative model may disagree with their human intuition and how to harness the model better in a downstream application.
>
> ***Formatting suggestions***
>
> **Q4. It is necessary to explain these datasets and experiments at the beginning of section 4**
> We thank the reviewer for their suggestion. We will modify the beginning of section to give a better outline of our experiments.
>
>
> **Q5. Table 3 is unclear with respect to the same source or same destination setting**
>
> In Table 3 we seek to answer questions Q1-Q3 and as a result, this experiment is neither in the same source nor the same destination setting but instead a different experiment with the goal of validating the applicability of our proposed CAGE framework.
>
>
> **Q6. The target image used of source manipulation**
>
> We thank the reviewer for bringing this to our attention. We have updated the manuscript based on the reviewers suggestion.
>
> **Q7. Given that the other places are written as 'Figure x.' …**
>
> We thank the reviewer for their suggestion, the manuscript is updated.
>
> **Q8. Two candidate options that act directly on the counterfactual latent are proposed in subsection 3.4. I am not quite sure which option is adopted in the calculation.**
>
> We are unsure what the reviewer means in their question. Our CAGE framework requires the calculation of a $\Delta \tau$ which means we must calculate both $m_c \to m_e$ and $m_e \to m_c$ directions. Note that the assignment of cause and effect is arbitrary and via $\Delta \tau$ we can ascertain which of the meta-attributes is truly a cause or an effect under the generative model. To assess independence we then perform a statistical test.

---

> > ### Comment · Reviewer_FwcQ · 2022-11-19
> > **Comments by Reviewer FwcQ**
> >
> > The authors have addressed some of the issues. However, the following issues still need to be further discussed.
> >
> > 1. Regarding capturing causal relationships among meta-attributes, I have the same concern as the Reviewer inuw. The proposed work cannot respect real-world cause-and-effect relationships. Therefore, the significance of the works should be carefully discussed. Otherwise, the motivation is too subjective.
> >
> > 2. To support the claim that some models may learn spurious causal structures, evidence should be experimentally or theoretically provided.
> >
> > 3. To make the reader better grasp the intuition, i.e., obtain fine-grained generation by using causal insights, it would be better to add illustrative examples in the Introduction.

---

> > > ### Author Response · Authors · 2022-11-25
> > > **Re: Rebuttal Response Part 1/3**
> > >
> > > **Q1. Regarding capturing causal relationships among meta-attributes … Otherwise, the motivation is too subjective.**
> > >
> > > We appreciate the reviewer's concern that whilst CAGE may uncover implicit causal relationships that are different from real-world causal mechanisms it is still a worthwhile endeavor. In particular, as we pointed out in our rebuttal to Reviewer inuw that the internal structure of any deep generative models has no guarantee of ever recovering—i.e. being identifiable—the true causal relationships that drive the data generation process. This is an important point to re-iterate, CAGE does not attempt to perform causal discovery on the data but rather on the internal representations learned by a model. We argue such an endeavor is objectively fruitful as we are able to extract key insights such as which meta-attribute to intervene on for conditional generation tasks. In section 4.1 we sought to categorize exemplar conditional generation applications (e.g. in painting, data augmentation) into either the same source or same destination setting and we empirically demonstrated that on the whole human evaluators agree that modifying the meta-attribute suggested by cage leads to higher quality samples in terms of visual fidelity. In the future, we also expect to see other downstream use cases of our framework with the growing fidelity of generated data and its consequent use as a substitute for real data, e.g., Jahanian et. al 2022 and Taori. & Hashimoto 2022 among many others.
> > >
> > > Part 1/3

---

> > > > ### Author Response · Authors · 2022-11-25
> > > > **Re: Rebuttal Response Part 2/3**
> > > >
> > > > **Q2. To support the claim that some models may learn spurious causal structures, evidence should be experimentally or theoretically provided.**
> > > >
> > > > We would like to communicate to the reviewer that there is indeed ample evidence (in our manuscript and beyond, for which we will provide references) of deep generative models learning spurious causal structures. Below, we start with a simple example involving 2 variables before commenting broadly on the larger theoretical results known in the community.
> > > >
> > > > Consider 2 variables, x and y, such that the true causal structure is x->y without loss of generality. For learning a generative model, we only observe samples from the joint p(x,y). Let’s assume we learn a masked autoregressive flow (MAF) model, which requires specifying an ordering. In this case, we can easily choose to learn either of the two orderings, $x \to y$ or $y \to x$, since there is no way to ascertain the true causal structure from the joint alone. We performed this experiment empirically as well where we learned two models, one each with the causal and anti-causal ordering. In both cases, we find that CAGE uncovers (---i.e. $\Delta \tau >0$) the ordering of the input variables which highlights how one can learn a good generative model whose internal causal representation may disagree with the true ground truth causal mechanism. We provide details of this experiment in our updated manuscript in appendix C.2
> > > >
> > > > More broadly, with regard to the theoretical aspect: a fundamental property through which to provide any form of guarantee that a model learns the true causal mechanism present in the data is identifiability. Identifiability, loosely speaking, implies that there is only one unique model parameterization that could have given rise to the observed data. A popular example of identifiability is the maximum likelihood estimator for exponential family models - there is only a single unique MLE parameter, therefore we can do inference over such a parameter.
> > > > The relationship between identifiability and causal inference has been studied for a long time: in the context of linear models, Shimizu et al (2006) demonstrated that only linear models with non-Gaussian latent disturbances are identifiable, thereby leveraging linear ICA algorithms to perform causal discovery.  The high-level idea behind this work was to study the estimated ICA unmixing matrix to infer causal structure - for further details, we refer the reviewer to Shimizu (2014).
> > > >
> > > > An important corollary of this work is that linear models with Gaussian latent variables (e.g., as in PCA) are not identifiable. This implies that a generative linear model, such as PCA, will encode arbitrary implicit causal relationships. To further demonstrate this, we note that the estimated PCA decomposition can be multiplied by an orthonormal matrix, R, and the solution continues to be optimal.  As a result, when any causal dataset is studied by PCA we cannot expect to infer causal structure by studying the PCA projection matrix. This is in sharp contrast to linear ICA, where the causal structure is encoded in the mixing matrix.  An alternative interpretation of the above theoretical results is that in the absence of identifiability, a generative model can encode arbitrarily (i.e., potentially spurious) causal associations.
> > > >
> > > > As can be expected, identifiability guarantees are even more difficult to obtain in the context of nonlinear models. However, there has recently been progress towards this goal in the work of Hyvarinen & Morioka (2017),  Hyvarinen et at (2019), and Khemakhem et al (2020). The models presented in those papers do achieve a similar flavor of identifiability over deep generative models, through which we can perform causal discovery as shown in e.g., Monti et al (2019). However, for such identifiability guarantees to hold we require strict assumptions on the data-generating process (e.g., latent variables must be piecewise stationary). Such assumptions are often not satisfied in practice, especially when training deep generative models, such as GANs, over large-scale image datasets. As a result, the analogous result from (linear) PCA holds in the context of (much deeper) generative models such as GANs: the implicit causal structure present within a generative model is arbitrary and need not agree with the true data-generating mechanism.
> > > >
> > > >
> > > > Part 2/3

---

> > > > > ### Author Response · Authors · 2022-11-25
> > > > > **Re: Rebuttal Response Part 3/3**
> > > > >
> > > > > **Q3. To make the reader better grasp the intuition, i.e., obtain fine-grained generation by using causal insights, it would be better to add illustrative examples in the Introduction.**
> > > > >
> > > > > We agree with the reviewer that the intuition for fine-grained generation may be hard to grasp and this is precisely the motivation for Figure 1. Succinctly, Figure 1 represents the same destination setting that is used in section 4.1 of the paper and it illustrates that manipulating the effect attribute (blonde hair) that CAGE uncovers (Gender -> Blonde hair) leads to a higher degree of visual fidelity when the goal is to conditionally generate males with blonde hair. We have updated the caption of this figure to highlight the usage of the term “same-destination” such that the reader is able to better grasp the intuition as well as understand the connection to our empirical validation in section 4.1. Also, we would like to point out that in section 1.1 of the introduction we use Figure 2 to illustrate the different settings of past work (e.g. CausalVAE) in comparison to CAGE which we believe aids in further building intuition. We hope that these figures are sufficiently clear to the reviewer. We are happy to modify them if the reviewer has any concrete intuitions that are not effectively communicated via these figures. Please, do let us know.
> > > > >
> > > > > Part 3/3
> > > > >
> > > > > **References**
> > > > > Jahanian, A., Puig, X., Tian, Y., & Isola, P. (2021). Generative models as a data source for multiview representation learning. arXiv preprint arXiv:2106.05258.
> > > > >
> > > > > Taori, R., & Hashimoto, T. B. (2022). Data Feedback Loops: Model-driven Amplification of Dataset Biases. arXiv preprint arXiv:2209.03942.
> > > > >
> > > > > Shimizu, S., Hoyer, P.O., Hyvärinen, A., Kerminen, A. and Jordan, M., 2006. A linear non-Gaussian acyclic model for causal discovery. Journal of Machine Learning Research, 7(10).
> > > > > Shimizu, S., 2014. LiNGAM: Non-Gaussian methods for estimating causal structures. Behaviormetrika, 41(1), pp.65-98.
> > > > >
> > > > > Hyvarinen, A., Sasaki, H. and Turner, R., 2019, April. Nonlinear ICA using auxiliary variables and generalized contrastive learning. In The 22nd International Conference on Artificial Intelligence and Statistics (pp. 859-868). PMLR.
> > > > >
> > > > > Hyvarinen, A. and Morioka, H., 2016. Unsupervised feature extraction by time-contrastive learning and nonlinear ica. Advances in neural information processing systems, 29.
> > > > >
> > > > > Khemakhem, I., Kingma, D., Monti, R. and Hyvarinen, A., 2020, June. Variational autoencoders and nonlinear ica: A unifying framework. In International Conference on Artificial Intelligence and Statistics (pp. 2207-2217). PMLR.

---

### Review · Reviewer_1ieq · 2022-10-23

**Summary Of Contributions:**

This paper propose a method called CAGE, which infers the cause-effect relationships within the latent space of deep generative models. It is done by geometric manipulation of latent vector with the help of a latent classifier. The authors evaluate the method with both synthetic toy data and high dimensional human face data. It is shown that CAGE can reliably extract correct cause-effect relationships in the setting where such a relationship is known, and in more complex setting such as CelebAHQ, the method can generate counterfactual examples to support the discovery of cause-effect relationships.

**Requested Changes:**

As I said above, the paper can be better if background knowledge of causal inference and more intuitive introduction of the method are added. In addition, related papers mentioned above should be discussed.

**Strengths And Weaknesses:**

Strength:

The paper tackles a problem that has not been studied extensively. Currently researchers lack a deep understanding of what samples can deep generative model generate and what concepts are learned, and this method is a good probe to these questions.

The idea is simple but effective. By moving latent variables across the plane defined by the classifier, counter factual samples can be effectively generated. Experiments on generating counter factual examples are impressive.

Weakness:

1. Insufficient background knowledge. For readers with little or no experience in causal inference, this paper is a bit hard to follow. It would be better to have some explanation of terminologies, such as counter factual, do operator, etc. Also, the method should be explained in a more intuitive way that audience from general ML background can understand.

2. Some recent related work in controllable generation and counterfactual generation is not discussed. In particular, with the help of a latent classifier, [1] propose to formulate an EBM on the latent space for controllable generation. In addition, [2] propose method for creating counterfactual examples.

[1] Controllable and Compositional Generation with Latent-Space Energy-Based Models, https://arxiv.org/abs/2110.10873
[2]  Counterfactual Generative Networks, https://arxiv.org/abs/2101.06046

---

> ### Author Response · Authors · 2022-10-31
> **Rebuttal**
>
> We thank the reviewer for their thoughtful comments and feedback. In particular, we appreciate that the reviewer feels that our paper tackles a problem that has not been studied extensively and that our method is a good “probe” to understand concepts learned by deep generative models. Furthermore, we are heartened to hear that the reviewer felt our idea was “simple but effective” and that our generated counter factual samples are “impressive”.  We now answer the main questions and concerns raised by the reviewer.
>
> **Insufficient background knowledge**
>
> We acknowledge the reviewers' concern that the accessibility of our paper could be improved with a more detailed treatment of causal inference as it pertains to our setting. Towards this end, we will add another appendix with more precise definitions of terminology such as the “do” operator, counterfactual, interventions, etc... by the end of the rebuttal period. We would also like to highlight that our current figures 2 and 3 serve to explain such terminology through intuitive visual aids and directly ties into how they are manifested in our setting.
>
> **Comparison with Related Work**
>
> We thank the reviewer for bringing these related papers to our attention. We will add further discussion in our related work section. Succinctly in [2] the authors directly define the SEM over shape, object texture, and background in order to generate counterfactuals. In contrast, this work is focused on uncovering the latent causal structure of a generative model. Meanwhile [1] also seeks to do controllable generation within a latent space using attribute classifiers but lacks any notion of causality between attributes which our work seeks to uncover and exploit for better controllable generation.

---

### Review · Reviewer_9whC · 2022-11-01

**Summary Of Contributions:**

In this paper, authors propose to discover how deep latent models "causally reason"  two conceptual features. They measure the causal reasoning direction by the cause effect direction between the two features. Since the target is the reasoning of NN but not the real data generation process,  counterfactual data can be created by manipulating the latent variables with the help of classifiers of the two features over the latent space. Then, we overpass the challenge in standard causal inference tasks. Afterwards, the authors design the detailed implementation and call this algorithm GATE. The statistical significance can be tested by the p-value of GATE under the null. Finally, authors leverage the causal direction to controlly generate images. The results are subjectively more realistic.

**Requested Changes:**

Please see my suggestions above. I expect the authors can respond to at least the first two items.

**Strengths And Weaknesses:**

Strengths:
1. The idea to detect the reasoning process of deep generative models is very interesting. It provides a new possibility that how causal inference (mainly used to help human make decision) can help AI.
2. The methodology is clean and reasonable, for the defined task.
3. The comparison of three approaches in Figure 2 is insightful.

Weaknesses:

My main concern is on the "controlable generation" part, which shows how GATE can help us on particular AI tasks. Intuitively, if we know the NN is reasoning along certain directions, then we'd better control the generation process by manipulating the cause. However, the current sec 4.1 fails to demonstrate this. Some suggestions: 1) explicitly define the generation tasks for both "same source" and "same destination manipulation" setting by explaining what are the two features and the possible control pipelines; 2) show the generative images if we do not manipulate along the reasoning directions and make quantitative comparisons; 3) discuss how the accuracy of classifiers influence the generation and the insights; 4) conduct experiments for different kinds of deep generative models and discuss the insights.

---

> ### Author Response · Authors · 2022-11-03
> **Rebuttal**
>
> We thank the reviewer for their positive view of our work! We are especially excited that the reviewer noted that our proposed methodology is “very interesting” as well as providing a “new possibility [for] causal inference to help AI” - this was exactly what motivated us to complete this work. We also thank the reviewer for their suggestions, which we respond to below:
>
> **Explicitly define the “same source” and “same destination” tasks**
>
> We thank the reviewer for highlighting potential confusion regarding the tasks described in section 4.1. We have added a new Figure 7 (in the appendix due to space constraints) where we further clarify the experimental setup.
>
>
> **Generative images when we do not manipulate along the reasoning directions**
>
> We are not sure what the reviewer is referring to here. We note that in Section 4.1, we indeed consider manipulations over both the cause and effect variable and report subjective differences (as measured by human annotators). Perhaps the reviewer was referring to the generative images along a random direction? We note that whilst we do not present generated samples under these projections in the manuscript, they are considered as part of the permutation test whose results are reported in Table 3. On page 8 we comment that our randomized permutation test considers samples after  “performing interventions over the latent space of G which are effectively random projections”.
>
>
> **Relationship between linear separability and sample quality**
>
> The reviewer correctly notes that there is a relationship between the classification accuracy---which we report for meta-attributes in Table 1---and the quality of generated samples (as well as the extent to which the generator can be “controlled”). This is to be expected as the proposed methodology relies upon the assumption of an “accurate and well-calibrated (ideally Bayes optimal) classifier” as detailed in Section 3.4 (bottom of page 7) and discussed in further detail in Section “Q2. Linear Separability”.
>
>
> **Experiments over different generative models**
>
> We note that in our current manuscript we have presented results over both GANs as well as Normalizing Flows; see MorphoMNIST experiments in section 4 related to the work of Pawlowski et al., (2020).  These represent two different families of generative models (implicit vs. likelihood-based) and our framework CAGE applies to both in equal measure. We agree with the reviewer that it would be interesting to explore and identify the implicit causal structure for other latent variable deep generative models and leave that to follow-up works.
>
>
> Moreover, the objective of our work, as noted by the reviewer, is to propose a novel methodology through which to leverage causal inference to better understand and leverage generative models---while we are certainly excited to explore findings of CAGE on a wide range of deep generative models, we leave that to future work.

---

### Decision · Action_Editors · 2022-12-01

**Recommendation:** Accept with minor revision

**Comment:**

The paper addresses an interesting question on how generative models can be made more interpretable and controllable with causal inference. All reviewers agree the problem to be interesting and worth investigating. At the same time, the reviewers have raised many questions, most of which focus on the clarity and motivation. The authors provide detailed explanations on all the questions, which I believe have sufficiently addressed the issues. In the revision, I would expect the authors to take the comments from all the reviewers and incorporate them to revise and improve the current version. This will make the paper more clear and stronger.

**Audience:**

Yes, the generative model community can find the paper interesting and useful because there is a little research in this topic.

**Claims And Evidence:**

This paper aims at making pretrained generative models more controllable by imposing causal reasoning in the latent space. Evidences are provided by extensive experiments results, which I believe have sufficiently supported the claims.